# SPECULATIVE ACTIONS: A LOSSLESS FRAMEWORK FOR FASTER AGENTIC SYSTEMS

**Naimeng Ye**, **Arnav Ahuja**, **Georgios Liargkovas**, **Yunan Lu**, **Kostis Kaffes, Tianyi Peng**
Columbia University
New York, New York, USA
{ny2336, aa5790, gl2902, yl4021, kk3664, tp2845}@columbia.edu

## ABSTRACT

AI agents are increasingly deployed in complex, interactive environments, yet their runtime remains a major bottleneck for training, evaluation, and real-world use. Typical agent behavior unfolds sequentially, where each action requires an API call that can incur substantial latency. For example, a game of chess between two state-of-the-art agents can take hours. We introduce *speculative actions*, a lossless acceleration framework for general agentic systems. Inspired by speculative execution in microprocessors and speculative decoding in LLM inference, our method uses faster models to predict likely future actions and executes them in parallel, committing only when predictions match. We evaluate speculative actions across gaming, e-commerce, and web search environments, and additionally study a lossy extension in an operating systems setting. Across domains, we achieve up to 55% next-action prediction accuracy, translating into substantial latency reductions. Finally, we present a cost–latency analysis that formalizes the tradeoff between speculative breadth and time savings. This analysis enables principled tuning and selective branch launching, to ensure multi-branch speculation delivers practical speedups without prohibitive cost growth.

## 1 INTRODUCTION

Large language model (LLM)-driven agents are shifting from single-shot predictions to processes that run inside rich environments: browsers, operating systems, game engines, e-commerce stacks, and human workflows. These environments are not incidental; they determine what the agent can observe and do, gate progress through interfaces and rate limits, and dominate end-to-end latency. In practice, agent behavior unfolds as a sequence of environment steps (tool calls, Model Context Protocol (MCP) server requests, human-in-the-loop queries, and further LLM invocations), each with non-trivial round-trip time and cost. As capabilities improve, a new bottleneck emerges: time-to-action in the environment. Even when accuracy is high, an agent that pauses too long between steps is impractical for interactive use or high-throughput automation.

| **OS Tasks** (Abhyankar et al., 2025) | **Deep Research** (OpenAI, 2025) | **Data Pipeline** (Jin et al., 2025) | **Kaggle Chess Game** (Kaggle, 2025) |
|:---:|:---:|:---:|:---:|
| 10–20 min | 5–30 min | 30–45 min | 1 hour |

Table 1: Estimated time state-of-the-art AI agents spend on various tasks/environments.

As shown in Table 1, AI agents may require tens of minutes to hours to complete a single run across different environments, a cost that grows significantly when hundreds or thousands of iterations are needed for reinforcement learning or prompt optimization (Agrawal et al., 2025).

This inefficiency arises from the inherently sequential nature of API calls. Thus, we ask a simple question in this paper:

*Must an agent interact with its environment in a strictly sequential manner?*

---

*Equal contribution

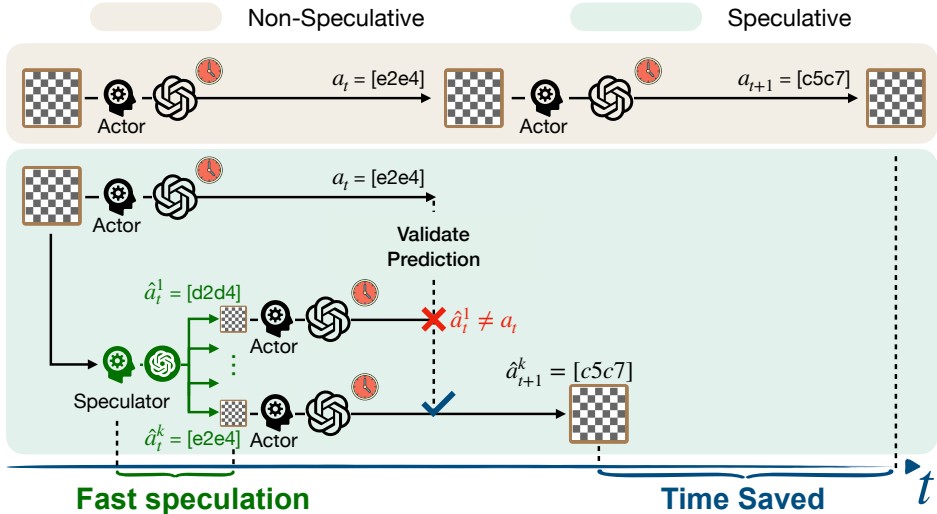

Figure 1: Illustration of our framework in a chess-playing environment. While the Actor issues an LLM call to decide the next move, the Speculator uses a faster model to guess it. These guesses enable parallel API calls for the next steps, and once a guess is verified, the system gains time through parallelization. The process runs in the backend, ensuring a lossless speedup for the user.

Our answer is no. Inspired by speculative execution in microprocessors and speculative decoding for LLM inference, we propose *speculative actions*: a general framework that allows agents to predict and tentatively pursue the most likely next actions using faster models, while slower ground-truth executors (powerful LLMs, external tools, or humans) catch up. In effect, the agent stages environment interactions (prefetching data, launching safe parallel calls, and preparing reversible side effects) so that validation, not waiting, is the critical path. When those slower evaluators confirm the guesses, progress has already been made; when they disagree, we execute as usual. The result is an *as-if-sequential, lossless interface with parallel, opportunistic internals*.

Concretely, in such agents, speculative actions introduce two roles in the environment loop:

- *Actor(s)*: authoritative but slow executors (e.g., SOTA LLMs, external APIs, environment's own responses, or humans) whose outputs materialize the ground truth for correctness and side effects.

- *Speculator(s)*: inexpensive, low-latency models that predict the next environment step, i.e., the action, its arguments, and the expected observation or state delta. Examples include smaller LLMs, same LLM with reduced prompts and reasoning steps, and domain heuristics.

A key design goal is losslessness relative to the environment's baseline semantics: speculative actions should not degrade final outcomes compared to a strictly sequential agent. We achieve this with (a) semantic guards (actors confirm equivalence of state transitions before commit), (b) safety envelopes (only idempotent, reversible, or sandboxed speculative side effects), and (c) repair paths (rollback or compensating actions when a guess is rejected). In many environments (e.g., web search, pre-checkout shopping carts, and OS-level operations in a sandbox) these patterns are natural and inexpensive to implement.

**Can we guess the next API calls of agents?** We show that, in practice, API intents can often be guessed with reasonable accuracy. In particular, we demonstrate speculative actions across four environments, each highlighting different aspects of agent latency:

- **Turn-based gameplay** (e.g., chess): the Speculator predicts the opponent's move while waiting for its turn. See Fig. 1.

- **E-commerce**: while conversing with a shopper, the Speculator proactively infers the shopper's intent (e.g., returning an item), and triggers tool calls in advance (e.g., checking return eligibility).

- **Multi-hop web search**: while awaiting results from slow external calls (e.g., Wikipedia), the Speculator can guess answers from its knowledge base, and execute subsequent search queries.

- **Operating systems (lossy extension)**: speculative, reversible actions react immediately to workload and environment changes, boosting end-to-end performance while actors confirm.

Across these settings, we observe substantially reduced latency, with up to 55% accuracy in predicting the next API calls and 20% end-to-end speedup. These results are achieved with a simple single-step speculation, and can be improved by advanced techniques such as adaptive speculation.

Finally, we give a cost-latency analysis that formally characterizes the tradeoff between speculating additional API calls and the resulting time savings. We provide a theoretical baseline for choosing the speculative breadth, and show that the cost incurred by confidence-based selection grows substantially slower than naively scaling the number of speculative branches. Furthermore, in our OS-tuning environment where losslessness is not required, cost and latency can actually both decrease. Our code is publicly available at `https://github.com/naimengye/speculative-action`.

### 1.1 RELATED WORK

**Speculative decoding and reasoning**  Our work is inspired by the use of speculative decoding in LLM inference. This technique accelerates autoregressive inference by using a small model to propose tokens which a larger target model verifies in batches, committing correct tokens and regenerating failures (Leviathan et al., 2023; Zhang et al., 2024; Chen et al., 2023). At the reasoning level, speculation has also been used to accelerate chain-of-thought (Wang et al., 2025b;a; Fu et al., 2025). Our framework adopts the same speculate-verify pattern at the level of API calls.

**Speculative planning for LLM agents**  More directly related are recent works on speculative planning for LLM-based agents (Hua et al., 2024; Guan et al., 2025). Hua et al. (2024) introduce interactive speculative planning, where a fast approximator proposes multi-step lookahead plans that are verified by a stronger model, with user interruption integrated. Their approach focuses on depth-oriented speculation along a single planning branch. Building on this, Guan et al. (2025) propose an online reinforcement learning method to dynamically determine the number of future steps to speculate, optimizing a cost-latency tradeoff while maintaining lossless execution.

Our work differs along two dimensions. First, we generalize speculation beyond planning to the *entire agentic environment*, including LLM calls, internal and external tool APIs, MCP-server interactions, and even human responses. This yields a unified framework for agentic speculation, particularly consistent with the emerging "environment" and MCP perspectives on agentic systems. Second, instead of depth-focused multi-step lookahead, we study a breadth-focused $k$-branch single-step strategy, where multiple actions are speculated in parallel at each step. We provide a cost-latency analysis for this scheme and derive closed-form expressions for expected time and token savings (Theorem 4). Section 5 compares breadth- and depth-focused strategies under a unified analytical framework. While Guan et al. (2025) optimize depth dynamically, we characterize the optimal number of speculative branches per step as a function of predicted accuracy (Section 5.2).

**Speculation in systems and architecture**  Speculation is prominently used in computer architecture to increase parallelism by executing instructions before their outcomes were resolved (Tomasulo, 1967) and rolling back when predictions were wrong (Lam & Wilson, 1992). In light of security vulnerabilities that exploit microarchitectural speculative execution, (Mambretti et al., 2019) developed Speculator to analyze CPU speculation behavior.

Similar ideas arise in systems software as thread-level speculation, which parallelizes sequential code under assumed independence and rolls back upon detecting data dependencies or conflicts (Estebanez et al., 2016). Recently, (Liargkovas et al., 2023) explored the use of tracing and containment to speculatively but safely run shell scripts out of order. Beyond traditional systems context, speculative techniques have also been used to parallelize otherwise sequential security checks (Nightingale et al., 2008),test configuration changes in isolation  (Su et al., 2007), and accelerate policy simulation in supply chain optimization (Farias et al., 2024).

## 2 FRAMEWORK

An agentic system is modeled as a Markov Decision Process (MDP) $(s_t, a_t)$, where $s_t$ denotes the state and $a_t$ the agent's action at step $t$. This model admits considerable flexibility: an action may represent a chatbot response, a tool call, or a button clicked by a computer-use agent, among others.

From a systems perspective, we model each action in an agentic system as an *API call*, which may block execution until a response is returned. This abstraction offers two key advantages: (1) it precisely defines what constitutes an action, and (2) it provides a unified framework for optimizing system latency, as we will see shortly. Notably, this perspective aligns with the recent development of MCP servers for agentic systems (Anthropic, 2024).

Formally, at each step $t$, the policy $\pi$ maps the current state $s_t$ to an API call:

$$(h_t, q_t) \leftarrow \pi(s_t),$$

where $h_t$ specifies the target API to invoke and $q_t$ its associated parameters. We write

$$\bar{a}_t \rightsquigarrow h_t(q_t) \qquad a_t \leftarrow \text{await}(\bar{a}_t)$$

to denote an asynchronous API invocation that returns a *future* (a pending action), and the await for when the response actually arrives. We use the bar notation (e.g., $\bar{a}$) for futures and a cache $C : (h, q) \mapsto \bar{a}$ that maps an API call specifier to its pending response. The left squiggly arrow indicates an asynchronous call with non-negligible delay.

The system subsequently transitions to the next state via a transition function $f: s_{t+1} \leftarrow f(s_t, a_t)$. As a concrete example, consider chess: the policy $\pi$ determines how to construct the prompt based on the current board state, $a_t$ corresponds to the move proposed by the LLM's response, and $f$ updates the board configuration accordingly. Note that the LLM call is the API, its *response* is the move $a_t$.

This formulation subsumes a broad range of realizations:

- **LLM calls:** each invocation of an LLM within the agent can be treated as an action.
- **Tool / MCP server calls:** each actual call for internal/external tools is treated as an action: e.g., terminal access, web search, deep research APIs, weather APIs, or browser-use MCPs.
- **Human-as-an-API calls:** furthermore, human responses themselves can be abstracted as API calls, often incurring even longer latencies than automated tools.

Given this abstraction, the fundamental bottleneck in executing agentic systems becomes apparent: each API call must complete before the next can be issued. To break this sequential dependency, we propose to **speculate a set of API responses** $\{\hat{a}_t\}$ using a faster model while waiting for the true response $a_t$. This enables speculative API calls for step $t + 1$ to be launched in parallel. At time $t$, if the API call $(h_t, q_t)$ can be found in the cache (cache hit), the system can skip the actual invocation and only wait for the pending action corresponding to this call to return (if not already returned). Formally, the algorithm is specified in Algorithm 1.

The resulting speedup relies on two key assumptions:

**Assumption 1** (Speculation accuracy). *The speculative model $\hat{g}$ guesses the current-step response $a_t$ accurately enough that the implied next call $(h_{t+1}, q_{t+1}) = \pi(f(s_t, \hat{a}_t))$ matches the true next call with probability $p > 0$.*

As shown later, this often holds in practice because API responses are typically predictable.

**Assumption 2** (Concurrent, reversible pre-launch). *Multiple API calls can be launched concurrently, and pre-launched calls that do not correspond to the realized trajectory have no externally visible side effects (or can be rolled back).*

In practice, this assumption is satisfied under modest traffic for many external APIs (e.g., web search, OpenAI LLM queries, email lookups). For self-hosted LLMs, concurrent calls also incur only minimal additional cost due to continuous batching.

We can then establish the following result (with proof deferred to the Appendix A).

**Proposition 1.** *Under Assumptions 1–2, suppose at each step the speculative branch implies the correct next call $(h_{t+1}, q_{t+1})$ with probability $p$, independently across $t \in [1, T-1]$. Let the latency of $\hat{g}$ be $\text{Exp}(\alpha)$ and the latency of the actual API call be $\text{Exp}(\beta)$ with $\beta < \alpha$. All latencies and guesses occur independently. Assume the transition $f$ and API parameter construction $\pi$ are negligible. Then the ratio between the expected runtime of Algorithm 1, denoted $\mathbb{E}[T_s]$, and the expected runtime of sequential execution, $\mathbb{E}[T_{seq}]$, is*

$$\frac{E[T_s]}{E[T_{seq}]} = 1 - \frac{1}{T} \frac{\alpha}{\alpha + \beta} \left[ \frac{(T-1)p(k)}{1 + p(k)} + \frac{p(k)^2}{(1 + p(k))^2} - \frac{p(k)^2}{(1 + p(k))^2}(-p(k))^{T-1} \right] \xrightarrow{T \to \infty} 1 - \frac{p(k)}{1 + p(k)} \cdot \frac{\alpha}{\alpha + \beta}$$

*where $p(k) = 1 - (1 - p)^k$ denotes the probability of at least one of the $k$ speculations hit.*

---

**Algorithm 1** Speculative actions with $k$-way parallel next calls

---

**Require:** Initial state $s_0$, horizon $T$, transition $f$, policy $\pi$, predictor $\hat{g}$, cache $C$. We use $\bar{a}$ to denote pending action.
1: **for** $t = 0$ to $T - 1$ **do**
2:      **Policy:** $(h_t, q_t) \leftarrow \pi(s_t)$
3:      **if** $(h_t, q_t) \in C$ **then**                                                  ▷ Cache hit
4:          $\bar{a}_t \leftarrow C[(h_t, q_t)]$
5:          $a_t \leftarrow \text{await}(\bar{a}_t)$                    ▷ Await pending action if not returned already
6:          $s_{t+1} \leftarrow f(s_t, a_t)$
7:          **continue**
8:      **end if**
9:      **Actor:** Issue real request (returns future): $\bar{a}_t \looparrowleft h_t(q_t)$
10:     **Speculator:** $\{\hat{a}_t^{(i)}\}_{i=1}^k \leftarrow \text{await}(\hat{g}(s_t, (h_t, q_t)))$       ▷ Actor and speculator run in parallel
11:     **for** $i = 1$ to $k$ **do**                                 ▷ One-step speculative rollout per guess
12:          $\hat{s}_{t+1}^{(i)} \leftarrow f(s_t, \hat{a}_t^{(i)})$
13:          $(\hat{h}_{t+1}^{(i)}, \hat{q}_{t+1}^{(i)}) \leftarrow \pi(\hat{s}_{t+1}^{(i)})$
14:          **Pre-launch**: $\bar{\hat{a}}_{t+1}^{(i)} \looparrowleft \hat{h}_{t+1}^{(i)}(\hat{q}_{t+1}^{(i)})$         ▷ Return pending action, hence non-blocking
15:                        $C[(\hat{h}_{t+1}^{(i)}, \hat{q}_{t+1}^{(i)})] \leftarrow \bar{\hat{a}}_{t+1}^{(i)}$          ▷ Cache speculative pending actions
16:     **end for**
17:     **Wait for resolved $a_t$ from Actor:** $a_t \leftarrow \text{await}(\bar{a}_t)$
18:     $s_{t+1} \leftarrow f(s_t, a_t)$
19: **end for**

---

Proposition 1 suggests the end-to-end latency reduction has an upper bound of 50%, occurring when $p = 1$ and $\alpha = \infty$. This can be further improved by the multi-step extension below.

**Extension** Algorithm 1 is only a simple demonstration of the idea. For example, one can naturally extend Algorithm 1 to *multi-step speculation*, where the Speculator predicts not only the next, but $s$ steps ahead. This yields a tree structure with deeper rollouts. This can be further combined with *adaptive speculation*: instead of generating $k$ guesses for $a_t$ uniformly, the Speculator also estimates confidence for each guess (e.g., via prompting LLMs or uncertainty-quantification methods), this is explored in Section 5. The most promising branches can then be expanded in a beam-search–like manner. Together, these ideas highlight the richness of speculative actions. Despite Algorithm 1's simplicity, the results from the four use cases in the following sections are already highly promising.

**Side effects and safety** Speculation executes a hypothesized next action $\hat{a}_{t+1}$ that may be wrong, so safety requires the ability to simulate first and then commit or roll back. In domains like chess, rollback is trivial; in others, overwrite is easy (e.g., OS tuning). But many systems involve irreversible or externally visible effects (e.g., deleting records, placing orders), where naive speculation is harmful. Thus, speculation must be limited to cases where mispredictions are reversible, via forking, snapshot restoration, or roll-forward repair (e.g., refund/replace).

## 3 ENVIRONMENTS

We now instantiate speculative actions in three environments—chess, e-commerce dialogue, and multi-hop web search—chosen to stress distinct latency bottlenecks (reasoning, tool/API round trips, and information retrieval). We pair a fast Speculator with a slow Actor and implement Algorithm 1.

### 3.1 CHESS ENVIRONMENT

We demonstrate the effectiveness of our framework in the context of multi-agent gameplay, focusing on chess as a canonical turn-based example. In standard play, analysis is strictly sequential: each player begins analysis only *after* the opponent has completed their turn. This serialization introduces substantial idle time. Particularly when both players rely on computationally intensive reasoning models, a single game can stretch to hours of wall-clock time. Our framework relaxes this constraint

through speculative parallel analysis, allowing players to anticipate and prepare for likely opponent moves in advance. We show that this results in significant reductions in overall game duration.

### 3.1.1 IMPLEMENTATION

We implement our framework on top of TextArena (Guertler et al., 2025), which provides a standardized gameplay interface for LLM-driven agents.

**Speculative pipeline**  At turn $t$, the game state $s_t$ corresponds to the current board position. The in-turn player issues an API call $h_t$ with parameter $q_t$ constructed from $s_t$ together with a reasoning-eliciting prompt. At this point, player $P$ is to move, and player $Q$ awaits. Proceeds as follows:

- Current in-turn player $P$: the player receives $s_t$, makes an API call $h_t$ to the agent with parameter $q_t = (s_t, prompt)$. This API call returns the next move $a_t = h(q_t)$, typically with high latency due to deep and extensive reasoning.

- Other out-of-turn player $Q$:

  1. **Prediction phase** The Speculator also receives the board state $s_t$ and issues an API call $\hat{h}_t$, using a prompt optimized for speed rather than depth. It returns the top-$k$ move predictions $\hat{a}_t^{(1)}, \hat{a}_t^{(2)}, \ldots, \hat{a}_t^{(k)}$, ordered by confidence.

  2. **Parallel computation** For each predicted move $\hat{a}_t^{(i)}$, the out-of-turn player $Q$ immediately launches a process analyzing a next move $\hat{a}_{t+1}^{(i)} = h_{t+1}(\hat{s}_{t+1}, prompt)$ for $i \in \{1, \ldots, k\}$, where $\hat{s}_{t+1}^{(i)} = f(s_t, \hat{a}_t^{(i)})$ denotes the next state resulting from applying the predicted action $\hat{a}_t^{(i)}$ to $s_t$.

  3. **Validation** When the current in-turn player $P$ finishes reasoning and returns its move $a_t$, we immediately check whether it matches any of the predicted moves $\hat{a}_t^{(1)}, \hat{a}_t^{(2)}, \ldots, \hat{a}_t^{(k)}$.

  4. **Commit or restart** If a match exists, we commit to the corresponding speculative branch, advancing directly to $s_{t+1} = f(s_t, a_t)$. The game thus skips ahead, terminating other threads. If no match exists, we discard all speculative branches and continue with $Q$'s regular move computation $a_{t+1} = h_{t+1}(f(s_t, a_t), prompt)$.

This pipeline is *lossless*: the final trajectory remains identical to non-speculative play, but time is saved through parallelized reasoning.

**Agent Configuration**  We find that using the same model for both Speculator and Actor, but with different prompts, maximizes prediction accuracy while keeping speculation fast. Accordingly, in our experiments, the Actor is instantiated with GPT-5 with high reasoning effort; and the Speculator is instantiated with GPT-5 configured with low reasoning effort and a specialized system prompt designed for rapid move prediction rather than exhaustive analysis.

### 3.1.2 RESULTS

We evaluate our framework in terms of both time saved and prediction accuracy. We track two metrics: (i) prediction accuracy: the fraction of rounds in which any speculative prediction matches the actual move; and (ii) time saved: $(T_{seq} - T_s)/T_{seq}$, where $T_s$ and $T_{seq}$ denote speculative and sequential execution times, respectively.

**More predictions improve time savings and accuracy.**  Figure 2 reports results over 30 steps. Our framework consistently reduces execution time, with larger savings as the number of speculative predictions increases. Across 5 runs, using 3 predictions yields an average time saving of 19.5% with an average prediction accuracy of 54.7%.

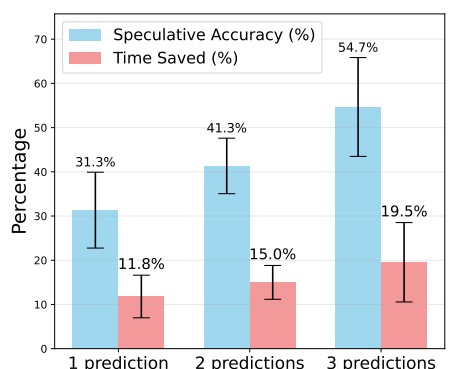

Figure 2: Percentage of time saved and percentage of correct predictions across 5 runs at 30 steps.

**Randomness of agent call in gameplay.** The variance in Figure 2 reflects realistic latency fluctuations from live API calls. Even with correct predictions, speedups vary: if the resulting position is trivial, little acceleration is realized; large gains occur only when predictions lead to positions requiring deep analysis. In addition, API latency itself is inherently stochastic. Backend load fluctuations (e.g., concurrent traffic to the model provider) can cause the same API call with different latency across runs. Consequently, measured latency reductions exhibit natural variability and are not perfectly reproducible.

## 3.2 E-Commerce Environment

Beyond competitive gameplay, customer-agent interactions in e-commerce provide a real-world setting where latency significantly impacts user experience. In a typical workflow, the customer submits a query through a chat interface and waits while the agent sequentially invokes multiple API calls—for example, processing a return may involve retrieving order information, validating eligibility for each item, and initiating the return. These chained calls can introduce substantial delay. By contrast, if some API calls are correctly speculated and executed in advance, the agent can return results immediately once the query arrives, making the interaction feel seamless. We evaluate this setting using the retail environment from $\tau$-bench (Yao et al., 2024).

### 3.2.1 Experimental Setup

**Speculative pipeline** In this scenario, the current state $s_t$ is defined as the conversation history up to turn $t$, and $h_t$ are the API calls required to answer the user's query (eg. get_user_details, get_order_details). Our Speculator will predict

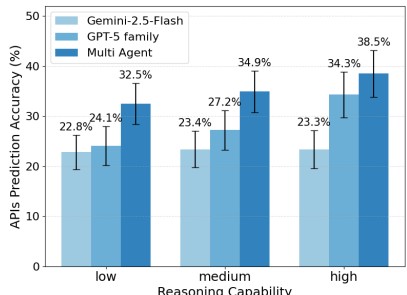

1. The user's query $\hat{a}_t$;

2. The target API calls and their corresponding parameters $(\hat{h}_{t+1}^{(i)}, \hat{q}_{t+1}^{(i)})$ for $i \in \{1, ..., k\}$, conditioned on the current state $s_t$ and the predicted user's query from step 1. Since the number of API calls for each turn is not fixed, the Speculator must also predict $k$.

Figure 3: APIs prediction accuracy across various Speculator models.

**Agent configuration** We evaluate multiple Speculator models, including OpenAI GPT variants (gpt-5-nano, gpt-5-mini, gpt-5) and Google Gemini (gemini-2.5-flash) under different reasoning budgets (1024/2048/4096 tokens). Motivated by prior work where heterogeneous LLM ensembles outperform single models (Jiang et al., 2023; Chen et al., 2025), we consider two configurations: (i) a *single-model* Speculator, and (ii) a *multi-model* Speculator, where comparable models run in parallel (e.g., gpt-5-nano with low-budget Gemini, gpt-5-mini with medium-budget Gemini). Their outputs are aggregated into a shared pool of candidate speculative actions.

At runtime, once the user simulator reveals the ground-truth utterance, the Actor validates the speculative API calls: correct predictions are committed immediately (eliminating latency), while incorrect ones are discarded without affecting correctness.

**Evaluation** We evaluate performance using **APIs prediction accuracy**, defined as the fraction of speculative API calls that match the ground-truth APIs required to resolve the user's query. This metric directly reflects the proportion of turns in which the user receives an immediate response, without waiting for API execution: higher prediction accuracy translates into greater time savings.

### 3.2.2 Results

Figure 3 shows that between 22% and 38% of API calls are correctly predicted by the Speculator. Accuracy improves with stronger models and the multi-agent configuration consistently outperforms single-model speculation. Importantly, low-budget models speculate in only 2–3 seconds (per the LLM API providers leaderboard[1]), well below the average user typing time of about 30 seconds (assuming 40 words per minute). This means that in roughly one third of turns, the agent can respond faster than sequential execution, without waiting for API execution.

---

[1]https://artificialanalysis.ai/leaderboards/providers

### 3.3 HotpotQA Environment

We further evaluate our framework on HotpotQA, a setting where the main performance bottleneck arises from information retrieval latency. In this example, the agent must answer multi-hop questions through sequential Wikipedia API calls (Yang et al., 2018), mirroring real-world agentic workflows with high round-trip network latency. In this setting, the Speculator predicts likely Wikipedia content while the actual API call executes. Parallelism allows the agent to continue reasoning on provisional information rather than blocking on API latency. See Appendix B.2 for details about our experimental setup.

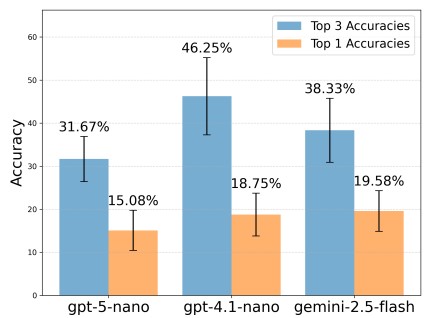

Figure 4: Accuracy with gemini-2.5-flash as the Actor. Speculating multiple actions ($k = 3$) yields higher accuracy than predicting a single action.

We evaluate on the accuracy of the predicted API calls. As shown in Figure 4, the Speculator successfully predicts ground truth API call up to 46% of the time with top-3 prediction. This accuracy improves significantly from top-1 to top-3 predictions, yielding substantial accuracy gains with modest speculation width increase. Our speculation provides value by precomputing reasoning paths during otherwise idle API waiting time.

## 4 Beyond Lossless Speculation: OS Hyperparameter Tuning Environment

Thus far, our experiments have focused on *lossless* speculation, where speculative actions are validated sequentially before commitment. We now turn to a *lossy* setting that relaxes this constraint. In latency-sensitive environments like an operating system, waiting for a powerful but slow Actor (10-15s deliberation) can leave the system in a degraded state. Instead, we use a fast Speculator to apply immediate provisional adjustments while the Actor deliberates. This is made safe by a last-write-wins mechanism—the Actor's final decision simply overwrites any speculative action, removing the need for complex rollbacks. This method accelerates convergence and improves reaction time, which we evaluate on the `sysbench cpu` benchmark, a CPU-bound workload (Kopytov, 2020).

### 4.1 Experimental Setup

We tune Linux's Completely Fair Scheduler (CFS) parameter `min_granularity`, which controls a task's minimum timeslice. This knob strongly affects scheduling performance: smaller timeslices reduce latency but can degrade throughput, yielding a classic trade-off. Building on (Liargkovas et al., 2025), we augment the prior LLM-based tuning setup with a speculative control loop.

The Speculator proposes a parameter update each second using the latest performance metric. The Actor, in contrast, responds every 10–15 seconds after analyzing a compressed chronology of the Speculator's recent (measurement, action) pairs. Upon arrival, the Actor's decision is applied immediately and its state resets the Speculator's context, preventing drift from the validated narrative.

**Evaluation**  We evaluate three systems: (1) **Actor-only**: slow but deliberative (10-15 s interval); (2) **Speculator-only**: fast (1 s interval) but non-extensive; (3) **Speculator–Actor**: combined system using speculative updates between Actor decisions.

### 4.2 Results

**Speculator mitigates poor-reaction slowdowns.**  As shown in Figure 5 (right), the Speculator significantly improves reaction time. During recovery, the full Speculator–Actor system maintains an average p95 latency of 37.93 ms, compared to 54.00 ms for Actor-only, which remains longer in degraded states (initially 102.97 ms). Fast speculative updates provide immediate mitigation while the Actor deliberates (details in §B.3.3).

**Speculator accelerates convergence to optimum.**  Figure 5 (left) shows that the joint system reaches the optimal setting (0.2 ms `min_granularity`) in 10-15 s, whereas Actor-only requires ~200 s and remains trapped in highly suboptimal regions (e.g., latency > 120ms) for extended periods. Rapid speculative exploration helps the Actor avoid pathological configurations.

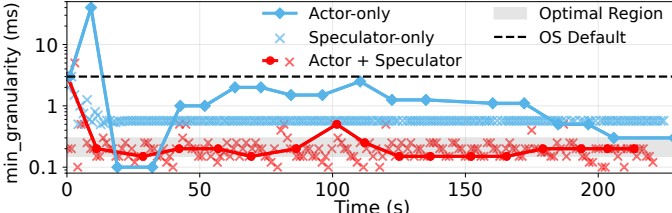

| Configuration | Latency p95 (ms) |
|---|---|
| Untuned | 102.97 |
| Actor-only | 54.00 |
| Actor + Spec. | 37.93 |

Figure 5: **(Left)** Comparison of **Speculator-Actor**, **Speculator-only**, and **Actor-only** convergence. The Speculator shortens time spent exploring poor settings. The Speculator-only agent stabilizes quickly but at a worse final value. **(Right)** Average p95 latency over a 20-second tuning experiment showing that rapid reaction offers immediate performance benefits (see §B.3.3). Lower is better.

**Speculator-only reacts quickly but is suboptimal.** While Speculator-only stabilizes rapidly, it converges to a worse configuration (0.55 ms; 36.24 ms latency) than the joint system (0.2 ms; 30.26 ms). Without the Actor's deeper reasoning, it cannot escape local minima.

**Cost and latency both decrease.** Despite additional speculative calls, total cost is lower due to faster convergence. As shown in Table 3, Actor-only converges at ∼200 s with a total cost of 2.18 cents, whereas Speculator–Actor converges in ∼13 s with only 0.17 cents.

Overall, the joint system combines fast adaptation with strategic guidance, achieving both responsiveness and optimal steady-state performance.

## 5   COST–LATENCY TRADEOFF

Performing more speculative API calls improves accuracy but also raises costs when pricing is based on the number of calls or tokens. In this section, assume a fixed token per unit time and fixed per token cost, and analyze the cost-latency tradeoff. Full details can be found in Appendix C.

### 5.1   BREADTH-FOCUSED SPECULATION (ALGORITHM 1)

In addition to Proposition 1, we obtain a closed-form expression for relative cost increase ratio

$$\lim_{T \to \infty} \frac{\mathbb{E}[M_{\text{spec}} - M_{\text{seq}}]}{\mathbb{E}[M_{\text{seq}}]} \le k - \left(k + \frac{\alpha}{\alpha + \beta}\right) \frac{p(k)}{1 + p(k)}.$$

See Theorem 4 in Appendix C for the formal theorem. Comparing Proposition 1 with the above expression, we see that both ratios are governed by $p(k)$. Thus, given an estimation of $p(k)$, a user can directly tune $k$ offline, trading off cost against latency. Our experiment (Figure 6) shows this non-linear dependence on $k$, and additional empirical results can be found in Appendix C.2.

### 5.2   DYNAMIC SELECTIVE SPECULATION.

So far we assume a fixed branch accuracy $p$. In practice, we sometimes are able to obtain per-speculation confidence estimates (e.g., from intrinsic model logits, or from a separately-trained auxiliary predictor), allowing confidence-aware selective speculation. At each speculation window, the accuracies of the $k$ speculative branches are random and drawn from a known distribution (which may vary over time). Before acting, the realized accuracy vector $\mathbf{p} = (p^1, \ldots, p^k)$ is observed, and we choose how many of the top branches to launch.

We model the cost-latency tradeoff via the weighted objective

$$\max \ r \cdot \sum_{t=1}^{T} \text{latency} - c \cdot \sum_{t=1}^{T} \text{cost}$$

where $r$ and $c$ encode the relative importance of latency and cost. For simplicity, let $a$ and $b$ denote the fixed latency of the actor and speculator, and define the latency gain $\ell = r(a - b) > 0$.

If the top $m$ branches are launched, the probability of a cached step is $q(m; \mathbf{p}) = 1 - \prod_{j=1}^{m}(1 - p^{(j)})$, where $p^{(1)} \geq \cdots \geq p^{(k)}$ are the sorted confidences.

**Theorem 3** (Confidence-aware selective speculation). *There exist scalars $\Delta_t$ such that at each speculation window t, the optimal breadth satisfies*

$$m_t^{\star}(\mathbf{p}) \in \arg \max_{m \in \{0,\ldots,k\}} \{q(m; \mathbf{p})\,\Delta_t - cm\}.$$

*The continuation values are given by the backward recursion*

$$\Delta_T = 0, \qquad \Delta_{1::T-1} = \ell - \mathbb{E}\Big[\max_m \{q(m; \mathbf{P})\,\Delta_{t+1} - cm\}\Big]$$

*In the stationary case (time-homogeneous accuracy distribution), the continuation values $\Delta_t$ collapse to a single constant $\Delta^{\star}$. Thus branches are added greedily in descending confidence order while $\Delta^{\star} \cdot \delta q(m; \mathbf{p}) \geq c$, where $\delta q(m; \mathbf{p})$ is the marginal gain from adding one more branch.*

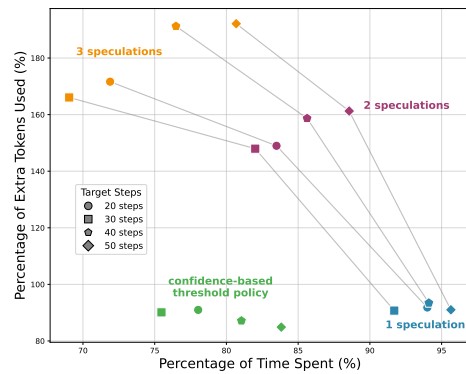

Figure 6: Cost-latency tradeoff across different speculation widths, forming a Pareto curve.

**Interpretation and implementation**  The key implication is structural: dynamic selection collapses to a one-dimensional trade-off. Additional branches are launched only when their incremental hit probability, scaled by a single continuation value ($\Delta_t$ or $\Delta^{\star}$), exceeds the marginal cost $c$. Under stationarity, this reduces to estimating $\Delta^{\star}$ offline; at runtime, the system simply sorts confidences and adds branches greedily, requiring $O(k)$ computation per step.

**Empirical results**  We implement a simple constant-threshold approximation of the stationary rule in the chess environment. At each step, after generating speculative branches, we use a predictor to estimate the correctness probability of each branch and continue only with those whose predicted accuracy exceeds 50%. This implements a simplified threshold rule consistent with the structure suggested by Theorem 3. Our method achieves the *lowest additional token cost* while providing *greater latency reduction* than naively launching 1 or 2 speculations per step.

### 5.3 Depth-focused speculation.

The previous two strategies are breadth-focused: each speculation is immediately followed by a real API call (speculative depth 1). Additionally, we analyze the opposite extreme: a *depth-focused* policy, in which multi-step speculations are continuously spawned. Somewhat counterintuitively, this strategy does *not* lead to exponential branch growth. Speculative calls are only extended when either a speculative or real call returns, and inconsistent subtrees are immediately pruned. Consequently, the system can run at most $a/b$ speculative steps ahead (governed by the relative speeds of real vs. speculative calls), ensuring that the number of active branches remains bounded and *does not scale with the horizon $T$*. Under this policy, we can show that (formal theorem in Appendix 6)

$$\frac{\mathbb{E}[T_{\text{seq}} - T_{\text{spec}}]}{\mathbb{E}[T_{\text{seq}}]} = \frac{T-1}{T}\,p\Big(1 - \frac{b}{a}\Big), \qquad \frac{\mathbb{E}[M_{\text{spec}} - M_{\text{seq}}]}{\mathbb{E}[M_{\text{seq}}]} \approx \frac{T-1}{T}\left((1-p)\left(\frac{a}{2b} - \frac{1}{2}\right) + p\right)$$

Compared to breadth speculation, depth speculation improves the latency coefficient from $\frac{p}{1+p}$ to $p$, increasing the theoretical speedup ceiling from $\frac{1}{2}$ to 1. The cost term scales with $(1 - p)(\frac{a}{2b} - \frac{1}{2})$, which captures how many speculative steps can accumulate before the real response arrives.

### 6 Conclusion

In this paper we propose **Speculative Actions**, a lossless framework for accelerating general agentic environments by breaking the strict sequentiality of their interaction loops. Our approach treats every step, whether an LLM call, tool invocation, MCP request, or human response, as an API call subject to prediction and parallelization. By pairing a fast *Speculator* with a slow but authoritative *Actor*, the framework enables agents to anticipate and prepare likely next actions in parallel, transforming otherwise idle waiting time into productive computation. We instantiate the framework across four representative environments and observe consistent substantial latency reduction. Finally, we provide a cost-latency analysis that addresses the tradeoff between the additional cost and the latency gains from launching additional speculative actions.

ACKNOWLEDGMENTS

This work is supported by Columbia-Dream Sports AI Innovation Center

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

# A   PROOF OF PROPOSITION 1

*Proof.* **Baseline.** In sequential execution, each of the $T$ steps requires one call to the true model $h$ with mean latency $1/\beta$. Therefore

$$E[T_{\text{seq}}] \;=\; \frac{T}{\beta}.$$

**Expected time saved per hit.** Consider two consecutive steps $(t, t+1)$. In the baseline, the total completion time is $R = B + C$, where $B, C \sim \text{Exp}(\beta)$ are the latencies of step $t$ and step $t+1$. With speculation, we launch $A \sim \text{Exp}(\alpha)$ during step $t$. If the guess is correct, the $(t+1)$ call $C$ can be issued once either $A$ or $B$ finishes, so the block completes at

$$S \;=\; C + \min\{A, B\}.$$

Thus, when a guess is correct, our expected time saved is

$$R - S \;=\; (B - A)_+,$$

where $(x)_+ = \max\{x, 0\}$.

By independence of $A, B$,

$$\mathbb{E}[(B - A)_+] = \int_0^\infty \int_0^b (b - a)\, \alpha e^{-\alpha a} \beta e^{-\beta b} \, da \, db = \frac{\alpha}{\beta(\alpha + \beta)}.$$

**Expected number of hits**. We denote the expected number of hits by round $n$ as $S_n$, that is

$$\mathbb{E}[\text{number of hits by round } n] = S_n$$

We then have $S_0 = 0$, $S_1 = p(k)$. In round 1, either (i) we hit with probability $p(k)$, in which case round 2 cannot be a hit (there is no speculation window immediately after a correct speculation), contributing $1 + S_{n-2}$; or (ii) we miss with probability $1 - p(k)$, after which round 2 proceeds normally, contributing $S_{n-1}$. We then have the following recursion

$$S_n = p(k)(1 + S_{n-2}) + (1 - p(k))S_{n-1}$$

Solve this linear recurrence by splitting into homogeneous and particular parts.

*(1) Homogeneous part*

$$S_n^h = p(k)S_{n-2} + (1 - p(k))S_{n-1}$$

The characteristic equation is

$$r^2 - (1 - p(k))r - p(k) = 0 = (r - 1)(r + p(k)) \implies \text{the roots are } r_1 = 1, r_2 = -p(k)$$

Therefore,

$$S_n^{(h)} = C_1 + C_2(-p(k))^n.$$

*(2) Particular solution* The forcing term is constant $(+p(k))$, and $r = 1$ is a root, so a constant trial collides with the homogeneous part. Try $S_n^{(p(k))} = an$ and substitute:

$$an = (1 - p(k))a(n - 1) + p(k)a(n - 2) + p(k) = an - a(1 + p(k)) + p(k),$$

which gives $a(1 + p(k)) = p(k)$ and thus

$$S_n^{(p(k))} = \frac{p(k)}{1 + p(k)}\, n.$$

Combine:

$$S_n = C_1 + C_2(-p(k))^n + \frac{p(k)}{1 + p(k)}\, n.$$

Use $S_0 = 0$ and $S_1 = p(k)$:

$$0 = C_1 + C_2, \qquad p(k) = C_1 + C_2(-p(k)) + \frac{p(k)}{1 + p(k)}.$$

Solving yields $C_2 = -\dfrac{p(k)^2}{(1 + p(k))^2}$ and $C_1 = \dfrac{p(k)^2}{(1 + p(k))^2}$.

Hence the closed form solution is

$$S_n = \frac{p(k)}{1 + p(k)} n + \frac{p(k)^2}{(1 + p(k))^2}\left(1 - (-p(k))^n\right)$$

**Total saving.** There are $T - 1$ potential speculation windows, hence

$$E[T_s] = \frac{T}{\beta} - S_{T-1}\frac{\alpha}{\beta(\alpha + \beta)}$$

**Final ratio.** Dividing by $E[T_{\text{seq}}] = T/\beta$ gives

$$\frac{E[T_s]}{E[T_{\text{seq}}]} = 1 - \frac{1}{T}\frac{\alpha}{\alpha + \beta}\left[\frac{(T - 1)p(k)}{1 + p(k)} + \frac{p(k)^2}{(1 + p(k))^2} - \frac{p(k)^2}{(1 + p(k))^2}(-p(k))^{T-1}\right]$$

$\square$

Taking $T \rightarrow \infty$, we get exactly that the ratio converges to $1 - \frac{p(k)}{1+p(k)}\frac{\alpha}{\alpha+\beta}$

# B  ADDITIONAL ENVIRONMENT DETAILS

## B.1  ECOMMERCE

**$\tau$-bench:**  A benchmark designed for dynamic task-oriented dialogues between a user (simulated by language models) and an API-augmented agent. The benchmark spans two domains — retail and airline, with structured databases, domain-specific tools. We focus on the retail domain, where the agent assists users with operations such as canceling or modifying pending orders, initiating returns or exchanges, or providing product and order information. The benchmark defines 115 tasks with 15 APIs (7 write, 8 read-only).

## B.2  HOTPOTQA

### B.2.1  EXPERIMENTAL SETUP

We build our framework upon ReAct (Yao et al. (2023)), which interleaves chain-of-thought with tool use.

**Speculative Pipeline** In this scenario, the state $s_t$ consists of the entire history of reasoning traces and retrieved information (API responses). At each step, the Actor takes in the current state $s_t$, selects an API call $h_t \in \{\text{Search}(), \text{Lookup}(), \text{Finish}()\}$ and a corresponding parameter $q_t$, e.g. Search(entity). The call $h_t(q_t)$ returns a response $a_t$, typically providing information about the queried entity. Our speculative framework operates as follows:

1. Speculator predicts the API call response $\hat{a}_t^{(i)}$, yielding predicted next states $\hat{s}_{t+1}^{(i)} = f(s_t, \hat{a}_t^{(i)})$, $i \in \{1, \ldots, k\}$.

2. Based on the states, the Actor generates reasoning traces and subsequently determines the next API decision $(\hat{h}_{t+1}^{(i)}, \hat{q}_{t+1}^{(i)})$ for $i \in \{1, \ldots, k\}$.

**Evaluation** We evaluate the effectiveness of the speculative pipeline by the accuracy of the predicted API call decisions $(\hat{h}_{t+1}, \hat{q}_{t+1})$. Specifically, we compare the predicted call against the ground-truth call $(h_{t+1}, q_{t+1})$ obtained under the true response $a_t$. We employ a strict match criterion, counting a prediction as correct only when $\hat{h}_{t+1} = h_{t+1}$ and $\hat{q}_{t+1} = q_{t+1}$. This stringent criterion captures whether speculation enables meaningful progress, as even minor parameter differences (synonyms, word order) count as mismatches.

**Agent configuration** We evaluate speculative accuracy across three Speculator models: GPT-5-nano, GPT-4.1-nano and Gemini-2.5-flash. For each model, we measure the top-k prediction accuracy, with $k \in \{1, 3\}$.

### B.2.2  RESULTS

Figure 4 shows that our Speculator successfully predicts the ground truth API call up to 46% of the time with top-3 prediction, despite our strict matching criterion. This accuracy improves significantly from top-1 to top-3 predictions, demonstrating that modest increases in speculation width yield substantial accuracy gains. Our speculation provides value by precomputing reasoning paths during otherwise idle API waiting time.

**Model Patterns** We observe high variation in API decision across different Speculators. These are largely driven by phrasing discrepancies – some models phrase the calls concisely while some over-specify. Interestingly, stronger models often yield lower accuracy, as their more diverse and context-specific queries (e.g., "List of Nobel laureates in physics 1970s" vs. "1970s Nobel Prize Physics winners list") are penalized under strict matching. In contrast, weaker models tend to produce simpler, more predictable outputs.

### B.3 Operating System Tuning

#### B.3.1 Experimental Setup and Implementation Details

**System and Workload Configuration** All experiments were conducted on a dedicated machine with 2× Intel Xeon Silver 4114 10-core CPUs at 2.20 GHz, 192 GB DDR4 RAM, and a 1 TB NVMe SSD running Ubuntu 22.04 with Linux Kernel 5.15, hosted on Cloudlab (Duplyakin et al., 2019).

We run `sysbench cpu` (Kopytov, 2020), a CPU-bound benchmark that repeatedly calculates a large prime number sequence. The benchmark reports several performance metrics every second. We run sysbench on 16 concurrent threads pinned on two CPU cores.

**Tuner Implementation Details** The system consists of two agents, a fast Speculator and a slow Actor, which collaborate to minimize the p95 latency of the workload. At each step, the tuner proposes a new configuration, which is applied to the live system. Applying the proposed parameters is a near-instant operation.

**CFS Parameter Details** The Completely Fair Scheduler (CFS) is a CPU scheduler for Linux that aims to give every task a fair share of CPU time. It exposes various hyperparameters that allow administrators to adjust its behavior. We tuned `min_granularity_ns`, which enforces a minimum timeslice a task will receive. The prompt templates guided the agents to explore a range from 50,000 to 50,000,000 nanoseconds (0.05 ms to 50 ms). The default value on Kernel 5.15 is 3 ms. Lower values for this parameter are expected to increase responsiveness at the cost of higher context-switching overhead, while higher values improve throughput but can worsen latency.

**History Compression and Context Management** To manage context window limits and costs, we employ different context strategies for the combined system versus the baselines.

In the **Actor-only** and **Speculator-only** baselines, the agents receive the full, unsummarized history of all previous iterations. For the Actor-only baseline, the low frequency of interaction (once every 10-15s) means the context grows slowly, rendering compression unnecessary within the benchmark duration.

In the **Speculator+Actor** combined system, history is managed via distinct prompt structures. When the slower Actor is invoked, its prompt context contains a fully compressed summary of all actions taken during its deliberation window. Each action from the Speculator is listed as a concise (parameter, result) pair. In contrast, the faster Speculator receives a hybrid context: it sees the same compressed history from the last Actor cycle, supplemented by the full, verbose replies from its own most recent actions. This dual-context mechanism allows the Actor to analyze long-term trends from a compact summary, while the Speculator retains immediate, detailed context for its rapid, reactive decisions.

#### B.3.2 Prompt Engineering for Multi-Agent Optimization

The following are the prompt templates used to guide the two LLM agents.

---

**Initial System Prompt for Actor and Speculator**

You are a Linux kernel scheduler tuning expert with deep knowledge of the Completely Fair Scheduler (CFS).

MULTI-AGENT ROLE: You are part of a MULTI-AGENT System.
[**For Actor**] You are the Actor. Your role is to provide thoughtful, well-analyzed parameter recommendations. You work alongside a Speculator that explores the parameter space rapidly. You will receive accumulated results from multiple agent calls to perform deeper analysis and identify trends.

[**For Speculator**] You are the Speculator. Your role is to provide immediate, intuitive parameter recommendations for each window. You work alongside an Actor that performs deeper analysis.

Your goal is to MINIMIZE p95 latency for a CPU-bound workload. The workload performance metrics might be NOISY, so look for consistent trends across configurations.

Tunable CFS parameter:

- `min_granularity_ns`: Minimum time slice before preemption. Lower values increase responsiveness but also overhead. Higher values improve throughput but can worsen latency.

Parameter Range:

- `min_granularity_ns`: 50,000 to 50,000,000 nanoseconds

Performance data will be provided in future calls. Respond ONLY in the format shown below:

```
Analysis:  <Your one or two-sentence decision reasoning>
Config:  { "min_granularity_ns":  <int> }
```

---

**Update for Speculator**

[*Context includes the compressed history for calls 1-10 and the raw Speculator responses for iterations 11-18*]

CURRENT BEST: p95 latency=[value] at call #[value]

Latest Result for call #19:
Config: "min_granularity_ns": [value] → p95 latency=[value]

Please provide your analysis and the next configuration for iteration #20.

---

**Update for Actor**

[*Context includes the compressed history for calls 1-10*]

CURRENT BEST: p95 latency=[value] at call #[value]

RESULT for call #11 [SPECULATOR]: min_granularity_ns=[value] → p95 latency=[value]
RESULT for call #12 [SPECULATOR]: min_granularity_ns=[value] → p95 latency=[value]
...
RESULT for call #19 [SPECULATOR]: min_granularity_ns=[value] → p95 latency=[value]

Please provide your analysis of the trend and the next configuration for call #20.

---

**Sample Agent Response**

Analysis: The performance peaked at 300,000 ns, suggesting the optimal value is likely in that region. I will narrow the search around that peak.
Config: { "min_granularity_ns": 250000 }

### B.3.3 Speculative Reaction Time Benefits

To provide a targeted example of how speculation mitigates transient performance loss, we conducted a controlled experiment. In this scenario, the system is deliberately perturbed at time $t_0$ by setting the `min_granularity` parameter to a highly suboptimal value (10 ms). We then compare the system's recovery under two configurations: the Actor-Speculator system and an Actor-Only baseline, which replays only the actions proposed by the Actor from the full Actor-Speculator trace.

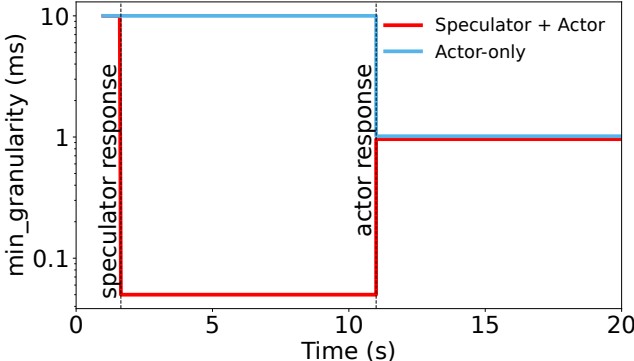

Figure 7: A controlled experiment showing the system's step response after a manual perturbation at $t = 0$. The **Actor-Speculator** system corrects the poor setting within a second, while the **Actor-only** system must wait over 10 seconds for its next decision cycle. The quantitative results of this experiment are summarized in Figure 5 (Right) in the main text.

As shown in Figure 7, the Actor-Speculator system reacts almost instantly. The fast Speculator, seeing the immediate performance degradation, applies a corrective action that brings the system back to an efficient state in about one second. In contrast, the Actor-Only system is forced to endure the poor performance for over 10 seconds, as it must wait for the slower Actor to complete its deliberation cycle before it can act. The performance gap shown in the plot is quantified in the main text (Figure 5, Right).

## C Cost–Latency Tradeoff

### C.1 Breadth-focused speculation details

Algorithm 1 launches $k$ parallel speculative branches at any step $t \in \{1, \ldots, T - 1\}$, which is then immediately followed with an API call. We assume each branch independently produces the correct next call with probability $p$. Let

$$p(k) := \Pr(\text{at least one branch is correct}) = 1 - (1 - p)^k.$$

We assume the latency of a speculative call is $\text{Exp}(\alpha)$, and the latency of a real API call is $\text{Exp}(\beta)$ with $\beta < \alpha$. Let $c$ denote the cost per unit time for both speculation and real API work. Let

$$\mathbb{E}[T_{\text{spec}}], \quad \mathbb{E}[M_{\text{spec}}]$$

denote the expected latency and cost under Algorithm 1 (1 step breadth speculation), and similarly let $(T_{\text{seq}}, M_{\text{seq}})$ denote the sequential process with no speculation.

**Theorem 4** (Cost–Latency Tradeoff for Breadth Speculation). *Under the setup above,*

$$\frac{E[T_{\text{seq}} - T_{\text{spec}}]}{E[T_{\text{seq}}]} = \frac{1}{T} \frac{\alpha}{\alpha + \beta} \left[ \frac{(T-1)p(k)}{1 + p(k)} + \frac{p(k)^2}{(1 + p(k))^2} - \frac{p(k)^2}{(1 + p(k))^2} (-p(k))^{T-1} \right]$$

$$\xrightarrow{T \to \infty} \frac{p(k)}{1 + p(k)} \cdot \frac{\alpha}{\alpha + \beta}$$

*For cost, letting $\tilde{k}$ denote the number of* distinct *actions produced across the $k$ speculative branches,*

$$\frac{\mathbb{E}[M_{\text{spec}} - M_{\text{seq}}]}{\mathbb{E}[M_{\text{seq}}]} = \tilde{k} - \frac{1}{T} \left( \tilde{k} + \frac{\alpha}{\alpha + \beta} \right) \left[ \frac{(T-1)p(k)}{1 + p(k)} + \frac{p(k)^2}{(1 + p(k))^2} - \frac{p(k)^2}{(1 + p(k))^2} (-p(k))^{T-1} \right]$$

$$\xrightarrow{T \to \infty} \tilde{k} - \left( \tilde{k} + \frac{\alpha}{\alpha + \beta} \right) \frac{p(k)}{1 + p(k)}.$$

*Note that $\tilde{k}$ is possibly different from $k$ because the $k$ independent speculations might have duplications, in which case we kill the duplicated speculation processes.*

*Proof.* The proof of the time savings ratio is given in Appendix A. For cost, we have

$$M_{seq} \propto \frac{T}{\beta}$$

$$M_{spec} \propto \frac{T}{\beta}(\tilde{k} + 1) - S_{T-1} \left( \tilde{k} \frac{1}{\beta} + \frac{\alpha}{\beta(\alpha + \beta)} \right)$$

where the speculative expression is due to each hit by time step $T - 1$ will result in (i) the cached next step not having a speculative window, hence does not launch any speculations (ii) over counting hit action's generation time with execution time.

Plug in expressions we obtained from Appendix A, we get the expression desired. □

### C.2 Additional Empirical Results

#### C.2.1 E-commerce

**Trade-off between Prediction Accuracy and Cost.** The time cost in Figure 8a consists of latency (Time to First Token) and output response time. The dashed vertical line represents the average user typing time, estimated at 40 words per minute. At this threshold, the multi-agent setting achieves approximately 34% prediction accuracy, meaning that in over one-third of cases the agent can return an immediate response without waiting for API execution. This demonstrates that speculation can transform user experience from perceptibly laggy to effectively real-time in tool-heavy environments.

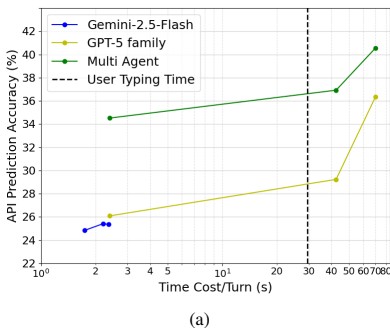 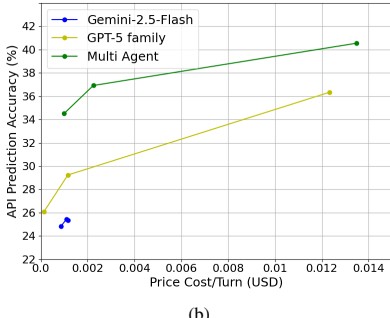

(a)                                    (b)

Figure 8: **Prediction Accuracy against Speculator's Cost across different models.** (a) Accuracy–Speculator time cost trade-off across models. The dashed line shows average user typing time. (d) Accuracy–Speculator price trade-off across models, reflecting the monetary cost of speculative execution.

### C.2.2 OS HYPERPARAMETER TUNING

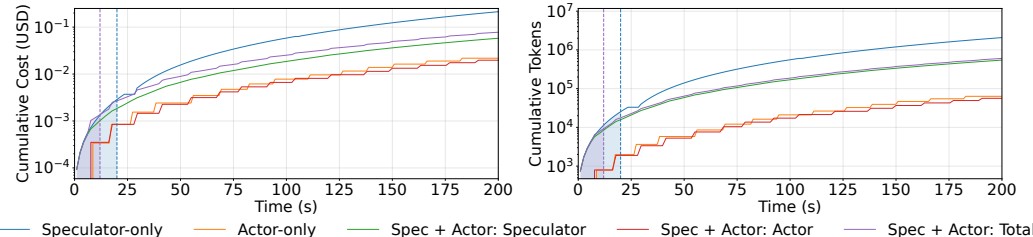

Figure 9: **Cumulative token usage and cost over time.** The left and right plots show the cumulative cost (USD) and total tokens used, respectively, for all three configurations. The vertical lines mark the observed convergence point for each system. The Actor-only model converges at 200s

Table 2: Cumulative tokens and cost (in cents) at selected time marks. While Speculation incurs higher instantaneous costs, its rapid convergence (bolded) prevents long-term resource waste compared to the slower Actor-only baseline.

| Elapsed Time | Actor-only | | Speculator-only | | Actor+Speculator (Total) | |
|---|---|---|---|---|---|---|
| | **Tokens** | **Cost (cents)** | **Tokens** | **Cost (cents)** | **Tokens** | **Cost (cents)** |
| Base Prompt | 744 | 0.02 | 690 | 0.01 | 1434 | 0.03 |
| **13s** | 1,216 | 0.05 | 14,973 | 0.18 | **12,135** | **0.17** |
| **20s** | 2,211 | 0.09 | **27,654** | **0.34** | 20,504 | 0.31 |
| 30s | 3,631 | 0.15 | 45,768 | 0.57 | 32,459 | 0.48 |
| 60s | 8,581 | 0.35 | 205,794 | 2.24 | 84,568 | 1.18 |
| 120s | 26,398 | 0.96 | 778,253 | 8.12 | 261,855 | 3.53 |
| **200s** | **63,376** | **2.18** | 2,099,894 | 21.5 | 607,877 | 7.83 |

Table 3

**Impact of Context Strategy on Cost.** While the Speculator operates at high frequency, the cost overhead is mitigated by the history compression mechanism described in §B.3.1. In the combined system, the expensive Actor model reads only a compressed summary of the Speculator's many steps, rather than the raw verbose logs. This keeps the prompt size for the Actor relatively stable compared to a linear growth of uncompressed history. As a result, the cost difference between the Actor-only and Speculator+Actor systems is driven primarily by the number of Speculator calls, rather than an explosion in context size per call.

As illustrated in Figure 9 and detailed in Table 2, the high frequency of the Speculator leads to rapid growth in token consumption and cost. In practice, however, this growth is bounded by the system's fast convergence. The combined Actor-Speculator system converges in approximately 13 seconds, while the Speculator-only system converges in 20 seconds. The Actor-only system converges after 200 seconds. Once an optimal state is reached, the tuning process concludes, rendering the potential for long-term exponential cost negligible in this context. Several optimization strategies, like truncating the context to a fixed window or disabling exploration after convergence, could further mitigate token growth but are left for future work.

## C.3 CONFIDENCE-AWARE SPECULATION

We formalize the branch selection problem introduced in Section 5.2.

**Model.** Fix a horizon $T$ and an integer $k \geq 1$. At each epoch $t$, the system is in mode $z_t \in \{0, 1\}$:

- $z_t = 0$: a speculation window is available,
- $z_t = 1$: a cached correct action must be executed (no speculation).

Let $a$ and $b$ denote the actor and speculator latencies, respectively, and define the latency gain

$$\ell := a - b > 0,$$

which is collected only at epochs with $z_t = 1$.

**Accuracy process.** At epochs with $z_t = 0$, an accuracy vector $\mathbf{P}_t = (P_t^1, \ldots, P_t^k) \in [0, 1]^k$ is realized and observed. Assume $\mathbf{P}_t \sim F_t$ independently across $t$, where the distribution $F_t$ may vary over time.

Given a realization $\mathbf{p}$, let $p^{(1)} \geq \cdots \geq p^{(k)}$ denote the sorted coordinates.

**Action and hit probability.** At $z_t = 0$, the agent chooses $m_t \in \{0, \ldots, k\}$, launching the top $m_t$ branches at cost $cm_t$. The probability of obtaining a cached action is

$$q(m; \mathbf{p}) = 1 - \prod_{j=1}^{m}(1 - p^{(j)}), \qquad q(0; \mathbf{p}) = 0.$$

**Mode transition.** If $z_t = 1$, then $z_{t+1} = 0$ deterministically. If $z_t = 0$ and action $m$ is chosen,

$$z_{t+1} = \begin{cases} 1, & \text{w.p. } q(m; \mathbf{P}_t), \\ 0, & \text{w.p. } 1 - q(m; \mathbf{P}_t). \end{cases}$$

**Reward.** The per-epoch reward is

$$R_t = \begin{cases} \ell, & z_t = 1, \\ -cm_t, & z_t = 0. \end{cases}$$

The objective is to maximize $\mathbb{E}\left[\sum_{t=1}^{T} R_t\right]$.

We then present the proof to the first part (general non-stationary) of Theorem 3.

*Proof.* Let $V_t^{(z)}$ denote the optimal expected total reward from epochs $t, \ldots, T$ given mode $z_t = z$, with terminal conditions $V_{T+1}^{(0)} = V_{T+1}^{(1)} = 0$.

By standard dynamic programming arguments, the Bellman equations are

$$V_t^{(1)} = \ell + V_{t+1}^{(0)}, \tag{1}$$

$$V_t^{(0)} = \mathbb{E}_{\mathbf{P} \sim F_t}\left[ \max_{m \in \{0, \ldots, k\}} \left\{ -cm + q(m; \mathbf{P}) V_{t+1}^{(1)} + (1 - q(m; \mathbf{P})) V_{t+1}^{(0)} \right\} \right]. \tag{2}$$

Define the continuation gap

$$\Delta_t := V_{t+1}^{(1)} - V_{t+1}^{(0)}.$$

Substituting equation 1 into equation 2 yields

$$V_t^{(0)} = V_{t+1}^{(0)} + \mathbb{E}_{\mathbf{P} \sim F_t} \Big[ \max_m \{q(m; \mathbf{P}) \Delta_t - cm\} \Big].$$

Hence, conditional on observing $\mathbf{p}$ at epoch $t$, the optimal decision maximizes

$$q(m; \mathbf{p}) \Delta_t - cm,$$

establishing the stated policy.

Finally, using

$$\Delta_t = V_{t+1}^{(1)} - V_{t+1}^{(0)} = \ell + V_{t+2}^{(0)} - V_{t+1}^{(0)},$$

and substituting the expression for $V_{t+1}^{(0)}$ gives the scalar recursion

$$\Delta_t = \ell - \mathbb{E}_{\mathbf{P} \sim F_{t+1}} \Big[ \max_m \{q(m; \mathbf{P}) \Delta_{t+1} - cm\} \Big],$$

with terminal condition $\Delta_T = 0$ (or equivalently, $\Delta_{T-1} = \ell$). $\qquad\square$

We then formally describe the stationary average reward corollary.

**Corollary 5** (Stationary infinite-horizon average-reward policy). *Assume the nonstationary model becomes stationary:*

- *$\mathbf{P}_t \sim F$ i.i.d. across $t$,*

- *the hit probability $q(m; \mathbf{p})$ and cost $c$ are time-invariant,*

- *the objective is to maximize long-run average reward.*

*Let $\bar{q}(m) := \mathbb{E}_{\mathbf{P} \sim F}[q(m; \mathbf{P})]$ denote the expected hit probability when launching $m$ branches.*

*Then the optimal average reward $g^\star$ satisfies*

$$g^\star = \max_{m \in \{0, \dots, k\}} \frac{\bar{q}(m)\, \ell - cm}{1 + \bar{q}(m)}.$$

*Define*

$$\Delta^\star := \ell - g^\star.$$

*Then an optimal stationary policy at a speculation window is*

$$m^\star(\mathbf{p}) \in \arg \max_{m \in \{0, \dots, k\}} \{q(m; \mathbf{p}) \Delta^\star - cm\}.$$

*In particular, the optimal policy mapping (the scalar $\Delta^\star$) is time homogeneous.*

*Proof.* Fix $\Delta^\star > 0$. Conditional on observing $\mathbf{p}$ at a speculation window, the stationary one-step objective is

$$f(m; \mathbf{p}) := q(m; \mathbf{p}) \Delta^\star - cm, \qquad m \in \{0, 1, \dots, k\}.$$

The first display in the corollary is exactly the definition of $m^\star(\mathbf{p})$ as any maximizer of $f(m; \mathbf{p})$.

To show the greedy marginal-threshold form, note that for sorted confidences $p^{(1)} \geq \dots \geq p^{(k)}$,

$$q(m; \mathbf{p}) = 1 - \prod_{j=1}^m (1 - p^{(j)}),$$

so the discrete marginal gain from adding the $(m + 1)$-th branch is

$$f(m + 1; \mathbf{p}) - f(m; \mathbf{p}) = \Delta^\star(q(m + 1; \mathbf{p}) - q(m; \mathbf{p})) - c = \Delta^\star \delta q(m; \mathbf{p}) - c,$$

where

$$\delta q(m; \mathbf{p}) := q(m + 1; \mathbf{p}) - q(m; \mathbf{p}) = \Big( \prod_{j=1}^m (1 - p^{(j)}) \Big) p^{(m+1)}.$$

Moreover, $\delta q(m; \mathbf{p})$ is nonincreasing in $m$ (diminishing returns), since $p^{(m+1)}$ is nonincreasing and the prefactor $\prod_{j=1}^{m}(1 - p^{(j)})$ is nonincreasing in $m$. Hence the increments $f(m + 1; \mathbf{p}) - f(m; \mathbf{p})$ are nonincreasing in $m$ as well.

Therefore there exists an index $m^{\star}(\mathbf{p})$ such that $f(m + 1; \mathbf{p}) - f(m; \mathbf{p}) \geq 0$ for all $m < m^{\star}(\mathbf{p})$ and $f(m + 1; \mathbf{p}) - f(m; \mathbf{p}) \leq 0$ for all $m \geq m^{\star}(\mathbf{p})$. Equivalently,

$$\Delta^{\star} \cdot \delta q(m; \mathbf{p}) \geq c \ \text{ for } m < m^{\star}(\mathbf{p}), \qquad \Delta^{\star} \cdot \delta q(m; \mathbf{p}) \leq c \ \text{ for } m \geq m^{\star}(\mathbf{p}),$$

which is precisely the greedy stopping rule stated in the corollary. □

### C.4 Depth focused search

In the most general setting, a policy may choose both (i) how many parallel speculative branches to launch, and (ii) how "deep" to unroll each speculative branch before initiating a real API call. Analyzing the optimal policy in this full space is highly non-trivial due to the branching structure of the execution tree.

To build intuition, we analyze two extrema regimes: (1) a *breadth-focused* regime, where at each step we launch $k$ parallel speculations and immediately follow each with an API call, and (2) a *depth-focused* regime, where execution follows a single branch as far as speculation and real API calls allow. These two settings correspond to simplified extremes of the decision space, providing interpretable analytic characterizations of the cost–latency tradeoff.

We analyzed the first in the previous section, and now analyze the opposite extreme: a depth-focused strategy. Under this policy, whenever either a speculative or real API call returns, the system launches *one* new real call and *one* speculation on top of that branch. If the real API result is inconsistent with the corresponding speculative guess, all descendants of that speculation are discarded. Let $p$ be the per-step correctness probability of a speculative guess.

Assume for simplicity that the real API call latency is deterministically $a$ and a speculative call latency is $b < a$.

**Theorem 6** (Cost–Latency Tradeoff for Depth Speculation). *Let $p$ be the probability that a speculation is correct at each step. Then under the depth-focused policy described above,*

$$\frac{\mathbb{E}[T_{\text{seq}} - T_{\text{spec}}]}{\mathbb{E}[T_{\text{seq}}]} = \frac{T - 1}{T} \, p\left(1 - \frac{b}{a}\right),$$

*and the expected cost satisfies*

$$\frac{\mathbb{E}[M_{\text{spec}} - M_{\text{seq}}]}{\mathbb{E}[M_{\text{seq}}]} = \frac{T - 1}{T} \frac{(1 - p)\left[a\lfloor\frac{a}{b}\rfloor - b\frac{(1+\lfloor\frac{a}{b}\rfloor)\lfloor\frac{a}{b}\rfloor}{2}\right] + pb\lfloor\frac{a}{b}\rfloor}{a} \approx \frac{T - 1}{T}\left((1 - p)\left(\frac{a}{2b} - \frac{1}{2}\right) + p\right)$$

*Proof.* We directly calculate the expected time cost of speculation execution as follows

$$T_{spec} = a + \sum_{s=0}^{T-1}\binom{T - 1}{s}p^{s}(1 - p)^{T-1-s}[sb + (T - 1 - s)a]$$

Let

$$S \sim \text{Binomial}(T - 1, p),$$

so that

$$\mathbb{P}(S = s) = \binom{T - 1}{s}p^{s}(1 - p)^{T-1-s}, \quad s = 0, \ldots, T - 1.$$

Then

$$\sum_{s=0}^{T-1}\binom{T - 1}{s}p^{s}(1 - p)^{T-1-s}[sb + (T - 1 - s)a]$$
$$= \mathbb{E}[Sb + (T - 1 - S)a]$$
$$= b\,\mathbb{E}[S] + a\,\mathbb{E}[T - 1 - S].$$

Since
$$\mathbb{E}[S] = (T-1)p,$$

we obtain
$$b(T-1)p + a(T-1-(T-1)p) = a(T-1) + (T-1)p(b-a).$$

Therefore,
$$T_{spec} = a + \sum_{s=0}^{T-1} \binom{T-1}{s} p^s (1-p)^{T-1-s} [sb + (T-s)a] = aT + (T-1)p(b-a).$$

Directly plug in $T_{seq} = aT$, we get the expression desired.

For cost, we know that with $(1-p)$ probability, speculation is inconsistent with true action, and the amount of tokens spent is from the branches that were spawn off before the correct action returns, that is

$$\text{Token of this step} = a + (a-b) + (a-2b) + \cdots + \left(a - \lfloor \frac{a}{b} \rfloor b\right)$$

$$= a \cdot \left(\lfloor \frac{a}{b} \rfloor + 1\right) - \frac{\left(1 + \lfloor \frac{a}{b} \rfloor\right) \cdot \lfloor \frac{a}{b} \rfloor}{2} \cdot b$$

With probability $p$, speculation matches with true action, in which case

$$\text{Token of this step} = a + \lfloor \frac{a}{b} \rfloor b$$

Therefore

$$M_{spec} = a + (T-1)\left[p\left(a + \lfloor \frac{a}{b} \rfloor b\right) + (1-p)\left(a \cdot \left(\lfloor \frac{a}{b} \rfloor + 1\right) - \frac{\left(1 + \lfloor \frac{a}{b} \rfloor\right) \cdot \lfloor \frac{a}{b} \rfloor}{2} \cdot b\right)\right]$$

□

**Interpretation.** Note that compared to breadth speculation, Depth speculation produces speedups unbounded in $p$ coefficient (the coefficient in front for breadth is $\frac{1}{2}$ whereas for depth speculation this coefficient is 1). In terms of cost, depth speculation cost has highest order term $(1-p)(\frac{a}{2b} - \frac{1}{2})$. This is governed by how many speculations one can spawn off before the current action is returned.

