# OpenReview forum: "Speculative Actions: A Lossless Framework for Faster AI Agents"
_ICLR.cc/2026/Conference — ICLR 2026 Oral_

### Official Review · Reviewer_qQio · 2025-10-30

**Soundness:** 3
**Presentation:** 3
**Contribution:** 3
**Rating:** 6
**Confidence:** 3

**Summary:**

The authors propose a way to increase the speed of agent-environment interactions by leveraging speculative execution.

They call this framework "Speculative Actions". It is a lossless framework that predicts the K most likely next actions using fast models, enabling multiple steps to be executed in parallel (before the real next action of the slow model is obtained).

The authors evaluate it on multiple settings, showing that it yields noticeable speedups.

**Strengths:**

This paper addresses the very practical need of maximizing efficiency and throughput in agentic settings, where API calls can be costly and time-consuming.

The authors show that the proposed method works well on a diversity of practical settings.

The paper is mostly easy to follow and understand. The logical order of presentation makes sense.

The paper includes bar charts that visually illustate the advantage of the proposed method. I appreciate the error bars in Figure 2.

**Weaknesses:**

There are a few places where terms, acronyms, or notations are used before first being defined and explained. More on this below.

Page 1:

Define what the acronym "MCP" means ("Model Context Protocol") before it's first used. Explain what it means, either here or on page 2 (where it says "MCP servers for agentic systems...").

Page 2:

"while waiting results" -> "while waiting for results"

Page 3:

"with simple implementation" -> "with a simple implementation"

"in computer architecture" -> "in the field of computer architecture"

"wrong and" -> "wrong, and"

"an Markov" -> "a Markov"

"(MDP) (st, at), where st denotes" -> "(MDP). We let st denote..."

Page 4:

"a set of API responses {\hat{a}_t}" -> "a set of k API responses {\hat{a}_t^(i)}_{i=1}^k". This explains what k refers to before it's used later on.

You have not defined what "Exp" means in Exp(α) and Exp(β). Define the notation before it's first used, by explicitly stating that Exp(λ) means an exponential distribution with rate λ.

Page 5:

"speculation need to be" -> "speculation must be"

"via fork" -> "via forking"

"Consider a game at turn t, " -> "Consider a game at turn t: "

"reasoning eliciting" -> "reasoning-eliciting"

"prompt" should not be in italics inside math formulas Use \text{prompt} to avoid this.

"applying predicted" -> "applying the predicted"

Page 6:

"match" should not be in italics inside math formulas. Use \text{match} to avoid this.

"If there exist no match" -> "If no match exists"

"next turn where Q is in turn to" -> "next turn, where it is Q's turn to"

"play while time is" -> "play, while time is"

"and computational complexity" -> "and the computational complexity"

Page 7:

"agent need to" -> "the agent needs to"

"is the API calls... that are needed" -> "is the API call... that is needed"

Page 8:

"predicts API call" -> "predicts the API call"

Space missing after "yielding predicted states".

"ht + 1" the "t + 1" is not properly subscripted.

"k ∈ {1,3}" should be typeset as a formula.

"k = 3" should be typeset as a formula.

Page 9:

"Our evaluation shows that the Speculator:" -> "Our evaluation shows that the Speculator-Actor system:" ?

**Questions:**

Page 3:

By "regenerating failures", did you really mean "recovering from failures"?

Figure 2: What do the error bars show? Standard deviation? Standard error? A 95% confidence interval for the mean (computed with which method)?

What's the best way to set k in practice? Can this be done in an online manner?

Alternatively, could one automate the selection of k situationally (i.e., on a step-by-step basis)? For example, you could have a fast model that's been trained to predict the *thinking times* of the slow models and/or transition function, and that could allow you to budget the number of speculations appropriately.

Are error bars missing from Figures 3 and 4? Were these run for multiple trials?

---

> ### Author Response · Authors · 2025-11-21
>
> We thank the reviewer for their thoughtful comments and feedback! We have fixed the mentioned typos and notation issues, and we address the rest of the comments below.
>
> **Q1:** Page 3: By "regenerating failures", did you really mean "recovering from failures"?
>
> **A1:** Here we meant in speculative decoding, when the larger target model disagrees with the next token prediction made by the smaller draft model (a failure), the larger model will have to regenerate the correct next token itself, starting from the last accepted prefix.
>
> **Q2:** Error bars
>
> **A2:** Thank you for the question! The error bars represent the standard deviation across multiple runs of the experiment, and we have added these error bars for the other two figures and clarified them in the paper.

---

> ### Author Response · Authors · 2025-11-21
>
> **Q3:** Adaptively selecting $k$
>
> **A3:** We fully agree that this topic warrants deeper investigation, and we thank the reviewer for the excellent suggestion of dynamically selecting k! This idea directly inspired one of our new results. We now include both theoretical and empirical analyses on k-selecting to achieve a favorable cost–latency tradeoff. We summarize the main findings below and provide full details in Appendix C.
>
> ---
>
> ### Finding 1
> **For static breadth-focused speculation (Algorithm 1), we obtain closed-form expressions for relative latency improvement and cost increase ratios. Practitioners can tune the number of speculative branches $k$ *offline* by directly assessing this tradeoff**. The formal result is given in Theorem 3 in Appendix C. Informally, given $T_{seq}, T_{spec}, M_{seq}, M_{spec}$ as the time and token cost of sequential execution and speculative framework, we have
> $$\frac{E[T_{\mathrm{seq}} - T_{\mathrm{spec}}]}{E[T_{\mathrm{seq}}]}
>     \xrightarrow{T\to\infty}
>     \frac{p(k)}{1+p(k)}\cdot\frac{\alpha}{\alpha+\beta}$$
> \begin{align*} \frac{E[M_{\mathrm{spec}} - M_{\mathrm{seq}}]}{E[M_{\mathrm{seq}}]}
>     \xrightarrow{T\to\infty}
>     k-\left(k+\frac{\alpha}{\alpha+\beta}\right)
>     \frac{p(k)}{1+p(k)}.
> \end{align*}
> where $p(k) = 1-(1-p)^k$ is the success probability of at least one of the $k$ speculations. Both ratios are governed by $p(k)$. Hence given an estimation of $p(k)$, a user can directly use these expressions to tune $k$ as a knob trading off cost increase against latency decrease. **Interestingly, our experiment (Figure 10 in Appendix C) exactly shows this non-linear dependence on $k$ in the cost increase ratio,** as the $p(k)$ also increase with $k$ increase.

---

> ### Author Response · Authors · 2025-11-21
>
> ---
>
> ## Finding 2
> If one could additionally obtain an estimate of the per-speculation accuracies online, then we can dynamically select speculations to continue with through a provably optimal threshold policy, thus effectively selecting $k$. This result is given in Theorem 5 in Appendix C.
>
>   In fact, we **empirically (Figure 10, Appendix C) implemented a threshold policy in the chess environment using these per-speculation accuracy estimates and observed substantial improvements in the cost–latency tradeoff.** At each step, after generating speculative branches, we use a predictor to estimate the correctness probability of each branch and continue only with those whose predicted accuracy exceeds 50\%. Our method achieves the *lowest additional token cost* while providing *greater latency reduction* than naively launching 1 or 2 speculations per step. Moreover, its time-saved performance is on average no more than about $5\%$ lower than that of always launching 3 speculations.
>
>
> | **Speculations**        | **Metric**                 | 20 steps | 30 steps | 40 steps | 50 steps |
> |-----------------------------------|----------------------------|----------|----------|----------|----------|
> | 1 Speculation                     | Time Saved Percentage (%)  | 6.00     | 8.30     | 5.89     | 4.35     |
> |                                   | Extra Tokens Used (%)      | 91.82    | 90.73    | 93.45    | 91.00    |
> | 2 Speculations                    | Time Saved Percentage (%)  | 16.51    | 17.99    | 14.38    | 11.44    |
> |                                   | Extra Tokens Used (%)      | 148.98   | 147.96   | 158.69   | 161.26   |
> | 3 Speculations                    | Time Saved Percentage (%)  | 28.11    | 30.96    | 23.52    | 19.32    |
> |                                   | Extra Tokens Used (%)      | 171.64   | 166.06   | 191.26   | 192.13   |
> | Confidence-based Selective Launch | Time Saved Percentage (%)  | **21.97** | **24.53** | **18.95** | **16.18** |
> |                                   | Extra Tokens Used (%)      | **90.97** | **90.11** | **87.14** | **84.89** |
>
>
>
>
>
>
>   **Theoretically, we prove that this type of policy is not only effective but *optimal***. We formulate the cost–latency tradeoff as a weighted objective
>     \begin{align*}\max \; r\cdot\sum_{t=1}^T\mathrm{latency} + c\cdot\sum_{t=1}^T\mathrm{cost}\end{align*}
>     where $r$ and $c$ encode the relative importance of latency and cost. We prove that the optimal one-step decision rule is precisely a **threshold policy**: accept a speculative branch (continue unrolling it) if and only if its predicted correctness $p_t$ exceeds a computable threshold $p_t^\star$. Thus, the practical policy we evaluated empirically aligns exactly with the theoretically optimal strategy.
>
>   ---
>
>   **Theorem 2** (Closed-Form Threshold rule)
>   The optimal one-step decision is
>   $$
>   \text{Accept speculation at step } t
>   \quad\Longleftrightarrow\quad
>   p_t \ge p^\star = ca+p_t\cdot\left(rb+cb+V(t+2) - V(t+1)\right)
>   $$
>   where $V$ calculated recursively backward from $V(T)=0, V(T+1) = 0$.
>   Under the stationary approximation,
>   $$
>       p^\star = -\frac{c a}{(r+c)b - g^\star},\qquad g^\star =       \max \left\lbrace
>           (r+c)a,
>           \frac{r a + 2c a + p(r b + c b)}{1+p}
>       \right\rbrace.
>   $$
>
>   ---
>
>   Additionally, this threshold behaves intuitively: (1) if speculation becomes cheaper, $p^\star$ decreases: the system speculates more often; (2) if speculation is generally unreliable (small $p$), $p^\star$ increases: only high-confidence branches are worth accepting. At deployment, the policy is *lightweight*: the system compares a single observed value $p_t$ to a precomputed global threshold $p^\star$.

---

### Official Review · Reviewer_hwQA · 2025-11-01

**Soundness:** 2
**Presentation:** 2
**Contribution:** 3
**Rating:** 6
**Confidence:** 3

**Summary:**

The paper proposes Speculative Actions, a framework for accelerating AI agent–environment interactions by predicting future actions with a smaller, faster “Speculator” model, while the main “Actor” validates them asynchronously. The idea is to make agent execution more parallel and efficient, analogous to speculative decoding in LLMs or speculative execution in CPUs. The paper demonstrates this concept across multiple domains (chess, e-commerce, HotpotQA, and OS tuning) and claims consistent speedups.

Overall, the motivation is reasonable as reducing agent latency makes sense, especially in complex API-driven workflows. However, in many realistic agent scenarios, I don't think the latency bottleneck is as severe as claimed, and the proposed speculative mechanism may introduce new costs or practical issues that are not fully addressed.

**Strengths:**

- The paper identifies a clear, relevant problem: latency in sequential agent–environment interactions.

- The proposed speculative framework is simple and implementable, with clear lossless and lossy variants.

- Multi-domain experiments demonstrate feasibility and some measurable speedups.

- The topic (efficient agent execution) is timely and of practical interest.

**Weaknesses:**

The paper provides strong empirical results but lacks deeper theoretical justification for why the speculative framework remains “lossless” under all conditions. Moreover, the evaluation mainly focuses on latency gains without a detailed analysis of trade-offs in resource consumption or potential instability in large-scale multi-agent settings. I may raise my evaluation on this paper if the authors could provide better justification for this concern.

**Questions:**

- The paper claims “up to 30% end-to-end speedup.” What is the variance across environments, and how were these averages computed?
- Could speculative execution introduce hidden costs that offset real-world gains?
- In multi-step speculation, how do you control exponential growth in parallel branches?
- How reproducible are the speedups given that API latency for large models may not be consistent?

---

> ### Author Response · Authors · 2025-11-21
>
> We thank the reviewer for their valuable suggestions and comments, and address the concerns raised below.
>
> **Cost-latency analysis**
> The reviewer raises an important point about the cost-latency trade-off. We fully agree that this topic warrants further exploration. We are excited to share additional theoretical and empirical results that explore this trade-off.
>
> 1. **Static breadth-focused speculation (Algorithm 1) and selection of k.** For Algorithm 1, we obtain closed-form expressions for relative latency improvement ratio and relative cost increase ratio. These expressions allow practitioners to tune the number of speculative branches k *offline* by directly assessing this tradeoff.
> 1. **Dynamic branch selection substantially reduces cost.** If additionally, one could obtain an estimation of the per-speculation accuracies online, then we prove theoretically the optimal strategy is a threshold policy: continue only when the predicted accuracy of a speculative branch exceeds a computable threshold. Empirically, this policy performs strongly—on the chess environment, it **achieves the same speedups with 40% fewer tokens**.
> 1. **No exponential blowup in depth-focused speculation.** We analyze a depth-focused speculative policy supporting multi-step speculation. We show that the number of parallel branches remains bounded. We also provide a characterization of the resulting cost-latency tradeoff.
> 1. **Environments that can reduce both cost and latency.** Interestingly, cost and latency do not always trade off against each other. In our OS-tuning environment where losslessness is not required, the Speculator accelerates convergence to the optimum so effectively that the system reaches the solution **using only 8% of the original time and 20% of the original token cost**.
> 1. **No/low cost for self-serving deployments.** The discussions above measure cost in terms of *token count*, but in **self-hosted inference** the relevant cost is GPU-hours or fixed procurement costs—not per-token pricing. Under low-traffic conditions with continuous batching, speculative calls become **effectively free** because they utilize otherwise idle GPU compute bandwidth. Moreover, this opens up new directions in scheduling and workload balancing for optimizing latency and throughput in agentic LLM deployments.
>
> Below, we summarize our main findings and refer the reviewer to Appendix C for full details. We will incorporate these findings in the main text of our camera-ready submission, and we thank the reviewer once again for eliciting this discussion.

---

> ### Author Response · Authors · 2025-11-21
>
> ## Result 1
>
> **Static breadth-focused speculation (Algorithm 1): Theorem 3 in Appendix C.** For Algorithm 1, which launches $k$ one-step speculative branches in parallel at each step, we obtain expressions for the relative latency reduction and cost increase:
>
> ---
>
>   **Theorem 1** (Cost-Latency Tradeoff for Breadth Speculation)
>
>   Let the expected latency of a speculative call be
>   $a$ and that of a real API call be $b$, with both following independent exponential distributions. Let $(E[T_{\mathrm{spec}}], E[M_{\mathrm{spec}}])$
>   denote the expected latency and the token cost under Algorithm 1, and similarly let $(T_{\mathrm{seq}},M_{\mathrm{seq}})$ denote the sequential process with no speculation.
> \begin{align*}
>     \frac{E[T_{\mathrm{seq}} - T_{\mathrm{spec}}]}{E[T_{\mathrm{seq}}]}
>     \xrightarrow{T\to\infty}
>     \frac{p(k)}{1+p(k)}\cdot\frac{b}{a+b}
> \qquad
>     \frac{E[M_{\mathrm{spec}} - M_{\mathrm{seq}}]}{E[M_{\mathrm{seq}}]}
>     \xrightarrow{T\to\infty}
>     \frac{k}{1+p(k)}-\frac{b}{a+b}
>     \frac{p(k)}{1+p(k)}
> \end{align*}
>   where $p(k) = 1-(1-p)^k$ is the success probability of at least one of the $k$ speculations.
>
> ---
>
>   Both ratios are governed by $p(k)$. Hence given an estimation of $p(k)$, a user can directly use these expressions to tune $k$ as a knob trading off cost increase against latency decrease. **Interestingly, our experiment (Figure 10 in Appendix C) exactly shows this non-linear dependence on $k$ in the cost increase ratio,** as the $p(k)$ increases when $k$ increases.

---

> ### Author Response · Authors · 2025-11-21
>
> ## Result 2
> **Confidence-based dynamic selective speculation: Theorem 5 in Appendix C.** The above strategy assumes knowledge of a fixed baseline accuracy $p$ for all speculations. In settings where we can obtain per-speculation confidence estimates (e.g., from intrinsic model logits, or from a separately-trained auxiliary predictor), we can bring down the cost-latency trade-off even more.
>
>   **Empirically (Figure 10, Appendix C), we implemented a threshold policy in the chess environment and observed substantial improvements in the cost–latency tradeoff.** At each step, after generating speculative branches, we use a predictor to estimate the correctness probability of each branch and continue only with those whose predicted accuracy exceeds 50\%. Our method achieves the *lowest additional token cost* while providing *greater latency reduction* than naively launching 1 or 2 speculations per step.
>
> | **Number of Speculations**        | **Metric**                 | 20 steps | 30 steps | 40 steps | 50 steps |
> |-----------------------------------|----------------------------|----------|----------|----------|----------|
> | 1 Speculation                     | Time Saved Percentage (%)  | 6.00     | 8.30     | 5.89     | 4.35     |
> |                                   | Extra Tokens Used (%)      | 91.82    | 90.73    | 93.45    | 91.00    |
> | 2 Speculations                    | Time Saved Percentage (%)  | 16.51    | 17.99    | 14.38    | 11.44    |
> |                                   | Extra Tokens Used (%)      | 148.98   | 147.96   | 158.69   | 161.26   |
> | 3 Speculations                    | Time Saved Percentage (%)  | 28.11    | 30.96    | 23.52    | 19.32    |
> |                                   | Extra Tokens Used (%)      | 171.64   | 166.06   | 191.26   | 192.13   |
> | Confidence-based Selective Launch | Time Saved Percentage (%)  | **21.97** | **24.53** | **18.95** | **16.18** |
> |                                   | Extra Tokens Used (%)      | **90.97** | **90.11** | **87.14** | **84.89** |
>
>
>
>   **Theoretically, we prove that this type of policy is not only effective but *optimal***. We formulate the cost–latency tradeoff as a weighted objective
>   $$\max \; r\cdot\sum_{t=1}^T\mathrm{latency} + c\cdot\sum_{t=1}^T\mathrm{cost}$$
>   where $r$ and $c$ encode the relative importance of latency and cost. We prove that the optimal one-step decision rule is precisely a **threshold policy**: accept a speculative branch (continue unrolling it) if and only if its predicted correctness $p_t$ exceeds a computable threshold $p_t^\star$. Thus, the practical policy we evaluated empirically aligns exactly with the theoretically optimal strategy.
>
> ---
>
>   **Theorem 2** (Closed-Form Threshold rule)
>   The optimal one-step decision is
>   \begin{align*}
>   \text{Accept speculation at step } t
>   \quad\Longleftrightarrow\quad
>   p_t \ge p^\star = ca+p_t\cdot\left(rb+cb+V(t+2) - V(t+1)\right)
>   \end{align*}
>   where $V$ calculated recursively backward from $V(T)=0, V(T+1) = 0$.
>   Under the stationary approximation,
>   \begin{align*}
>       p^\star
>       =
>       -\frac{c a}{(r+c)b - g^\star},
>       \qquad
>       g^\star
>       =
>       \max\!\left\lbrace
>           (r+c)a,
>           \frac{r a + 2c a + p(r b + c b)}{1+p}
>       \right\rbrace.
>   \end{align*}
>
> ---
>
>   Additionally, this threshold behaves intuitively: (1) if speculation becomes cheaper, $p^\star$ decreases: the system speculates more often; (2) if speculation is generally unreliable (small $p$), $p^\star$ increases: only high-confidence branches are worth accepting. At deployment, the policy is *lightweight*: the system compares a single observed value $p_t$ to a precomputed global threshold $p^\star$.

---

> ### Author Response · Authors · 2025-11-21
>
> ## Result 3
>
> **Depth-focused speculation: Theorem 7 in Appendix C.** The previous two strategies are breadth-focused: each speculation is immediately followed by a real API call (speculative depth 1). Additionally, we also analyzed the opposite extreme: a *depth-focused* policy, in which multi-step speculations are continuously spawned. **Somewhat counterintuitively, this strategy does *not* lead to a unbounded exponential blowup in the number of parallel branches.** Under this policy, new speculative and real API calls are launched only when either a speculative call or the real Actor call returns. If the real API result is inconsistent with the corresponding speculative guess, all descendants of that speculation are discarded. Consequently, the system can run at most $a/b$ speculative steps ahead (governed by the relative speeds of real vs. speculative calls), ensuring that the number of active branches remains bounded and *does not scale with the horizon $T$*.
>
>   Formally, for a one-speculation-per-step depth policy, we obtain the following cost–latency characterization.
>
> ---
>
>   **Theorem 3** (Cost-Latency Tradeoff for Depth Speculation)
>   Let $p$ be the baseline correctness probability of a speculative guess. Let $a$ and $b$ denote the fixed latency of real API and each speculation for simplicity. Using the same $T$ and $M$ notation, we have
>   $$
>       \frac{E[T_{\mathrm{seq}} - T_{\mathrm{spec}}]}{E[T_{\mathrm{seq}}]}
>       = \frac{T-1}{T}  p \Bigl(1 - \frac{b}{a}\Bigr),\qquad
>       \frac{E[M_{\mathrm{spec}} - M_{\mathrm{seq}}]}{E[M_{\mathrm{seq}}]}
>       \approx\frac{T-1}{T}\left((1-p)\left(\frac{a}{2b}+\frac{1}{2}\right)+p\frac{b}{a}\right)
>   $$
>
> ---
>
>   This result highlights two contrasts with breadth speculation.
>   First, the latency-speedup coefficient improves from $\frac{p}{1+p}$ (breadth) to $p$ (depth), raising the theoretical upper limit from $\tfrac{1}{2}$ to $1$. Second, the key term in the cost ratio, $(1-p)(\tfrac{a}{2b}+\tfrac{1}{2})$, reflects how many speculative steps can be launched before the real Actor response arrives. In the limit of perfect speculative accuracy, the ratio simplifies to $b/a$, corresponding to the extra cost of attaching a single speculative call to each real action.

---

> ### Author Response · Authors · 2025-11-21
>
> ## Result 4
> **Lossy-extension environments.** In environments where strict "losslessness" is not required, we can deploy the Speculator as a fast explorer that accelerates convergence. Although this introduces higher *instantaneous* cost, the *total* cost is actually *lower* because the system reaches convergence significantly earlier than the Actor-only baseline. For example, in our OS hyperparameter tuning experiment, the Actor-only baseline converges at roughly $200\text{s}$ with a total cost of $2.18$ cents, whereas the Actor+Speculator method converges around $13\text{s}$ with a total cost of only $0.17$ cents.
>
> | **Strategy**        | **Convergence Time** | **Tokens** | **Cost (cents)** |
> |---------------------|-----------------------|------------|------------------|
> | Actor-only          | 200s                  | 63,376     | 2.18             |
> | Actor+Speculator    | 13s                   | 12,135     | 0.17             |

---

> ### Author Response · Authors · 2025-11-21
> **Other questions**
>
> ## Q1 Why the speculative framework is "lossless."
>
> **A1** Our framework assumes that the Actor and Speculator do not interfere with each other’s state—a condition that holds broadly in our applications (e.g., chess, e-commerce, HotpotQA). Under this assumption, **the framework is lossless by construction**: as defined in Algorithm 1, the Actor only commits cached correct API calls produced by the Speculator. Therefore, the final execution trace is guaranteed to be identical to the trace obtained when the Actor runs sequentially without speculation.
>
> One may consider a more general setting where the Speculator is allowed to mutate the environment state, potentially influencing the Actor. If a rollback mechanism is available, our analysis extends naturally to this case as well. Developing such stateful speculative systems is an interesting direction for future work.
>
> **We also want to emphasize that "lossless" execution is not always the desirable objective.** In our OS-tuning applications, for example, speculative execution achieves faster and better convergence than Actor-only execution. In this case, the trajectory is deliberately not lossless—indeed, it is better than the sequential Actor trajectory.
>
> ## Q2: Multi-agent stability.
>
> **A2:** The speculation framework extends naturally to general multi-agent settings, as such systems can be represented as sequential API calls within an underlying MDP—particularly in synchronous multi-agent environments where agents advance in lockstep.
>
> For asynchronous multi-agent systems, indeed additional engineering challenges arise, such as coordinating many concurrently spawned speculative processes and managing potential rollbacks. This setting is closely related to the Time Warp mechanisms used in distributed parallel discrete-event simulation [1]. Developing a full speculative framework for such coupled, asynchronous multi-agent environments is an interesting direction for future work.
>
> ## Q3: What is the variance of speedup across environments and how these averages are computed.
> **A3:** In the chess environment, the end-to-end latency reduction ranges from 15% to 30% across different experimental setups with $k=3$. The reported averages are computed over multiple independent runs, with the corresponding standard deviations shown in Figure 2. Similar variability for speculation accuracy is reported for other environments as well.
>
> ## Q4: Could speculative execution introduce hidden costs that offset real-world gains
> **A4:** Extra API calls issued by the Speculator may introduce overhead, but as detailed in our cost--latency tradeoff analysis, we propose several techniques to mitigate this overhead. Moreover, in certain environments (e.g., OS tuning), speculation can in fact reduce *both* cost and latency by achieving substantially faster convergence.
>
> We also acknowledge that there are scenarios where the additional cost may be nontrivial. However, a key advantage of our approach is that speculation can be implemented entirely as a backend feature: the frontend user observes only improved latency, while the server operator can selectively enable speculation when its benefits are significant (a practice similar to speculative decoding in real-world LLM systems).
>
> ## Q5: In multi-step speculation, how do you control exponential growth in parallel branches?
> **A5:** Perhaps surprisingly, our theory shows that even under multi-step speculation, the number of speculative branches is bounded by $k^{a/b}$, rather than $k^{T}$, because branches *collapse* whenever an action from the Actor is revealed (see the new Appendix C for details). Furthermore, this $k^{a/b}$ bound can be reduced in practice by adaptively selecting only the most promising branches, as demonstrated in our experiments.
>
> We note that obtaining an optimal strategy for multi-step speculation may require a fine-grained dynamic programming formulation, which we leave as an interesting direction for future work.
>
> ## Q6: How reproducible are the speedups given that API latency for large models may not be consistent?
> **A6:** We want to emphasize our experiments directly use *real* API calls, so any variability in API latency is already reflected in the reported results. We fully agree that both latency fluctuations and variation in the model outputs can affect runtime—indeed, this is evident in the error bars in our chess results. However, tasks like chess involve inherently long reasoning per call, making the *expected* latency much larger than the run-to-run variance. As a result, we consistently observe a substantial *baseline speedup*, with some runs simply saving more than others.
>
> **Reference.**
> 1. Fujimoto, Richard M. "Parallel discrete event simulation." Communications of the ACM 33, no. 10 (1990): 30-53.

---

### Official Review · Reviewer_SJo2 · 2025-11-03

**Soundness:** 4
**Presentation:** 4
**Contribution:** 4
**Rating:** 10
**Confidence:** 3

**Summary:**

The paper proposes the concept of speculative actions, which uses speculative models in sequential environments to achieve significant speedups (up to 30%). Concretely, the paper considers the setting where an API determines the next action based on the current state, and this API is instantiated by either an expensive LLM (e.g., high-reasoning mode) or a human response. The paper proposes an algorithm that uses a much faster model, in this case a smaller LLM, to predict the likely output and to precompute the next state based on this action. If the action matches the prediction of the expensive model (or human), the algorithm can directly proceed to the next state, thereby processing two steps at a time. Otherwise, the environment generates the next state based on the true action, progressing without overhead compared to the sequential baseline. The paper considers four different environments with different constraints, and shows that the proposed approach achieves speedups of up to 30%.

**Strengths:**

The paper considers an important topic, namely, decreasing the latency of LLM agents in sequential environments, and introduces a novel algorithm to achieve significant speedups. The proposed approach is naturally inspired by speculative decoding from other domains, but the paper makes a significant contribution by showing that it is also applicable to this setting. The paper is well written, with clear illustrations and a convincing motivation. The claims are backed up by substantial experimental evidence across a wide range of real-world environments.

**Weaknesses:**

I think the cost-vs-latency tradeoff is an important aspect of this approach and should be discussed in the main paper, e.g., using the additional page of the camera-ready version (if applicable). In a practical application, it would be great for the user to have a tunable knob between costs and latency.

I found two typos: sequantial (L293) and gameply (L316).

**Questions:**

How can a user find an appropriate speculative model? Did any smaller LLMs not achieve a satisfactory performance?

---

> ### Author Response · Authors · 2025-11-21
>
> We thank the reviewer for their thoughtful feedback and constructive suggestions, and are very grateful that they found our work well-motivated and well-executed.
>
> **Cost-latency analysis.**
> The reviewer raises an important point about the cost-latency trade-off. We fully agree that this topic warrants further exploration. We are excited to share additional theoretical and empirical results that explore this trade-off.
>
>
> 1. **Static breadth-focused speculation (Algorithm 1) and selection of k.** For Algorithm 1, we obtain closed-form expressions for relative latency improvement ratio and relative cost increase ratio. These expressions allow practitioners to tune the number of speculative branches k *offline* by directly assessing this tradeoff.
> 1. **Dynamic branch selection substantially reduces cost.** If additionally, one could obtain an estimation of the per-speculation accuracies online, then we prove theoretically the optimal strategy is a threshold policy: continue only when the predicted accuracy of a speculative branch exceeds a computable threshold. Empirically, this policy performs strongly—on the chess environment, it **achieves the same speedups with 40% fewer tokens**.
> 1. **No exponential blowup in depth-focused speculation..** We analyze a depth-focused speculative policy supporting multi-step speculation. We show that the number of parallel branches remains bounded. We also provide a characterization of the resulting cost-latency tradeoff.
> 1. **Environments that can reduce both cost and latency.** Interestingly, cost and latency do not always trade off against each other. In our OS-tuning environment where losslessness is not required, the Speculator accelerates convergence to the optimum so effectively that the system reaches the solution **using only 8% of the original time and 20% of the original token cost**.
>
> 1. **No/low cost for self-serving deployments.** The discussions above measure cost in terms of *token count*, but in **self-hosted inference** the relevant cost is GPU-hours or fixed procurement costs—not per-token pricing. Under low-traffic conditions with continuous batching, speculative calls become **effectively free** because they utilize otherwise idle GPU compute bandwidth. Moreover, this opens up new directions in scheduling and workload balancing for optimizing latency and throughput in agentic LLM deployments.
>
> We refer the reviewer to the response to hwQA for a detailed summary.
>
> **Choose a proper speculator.** The reviewer also asks how users can identify an appropriate speculative model. In practice, this is a **model-selection** problem, closely related to recent work on routing queries to the most suitable LLM [1,2]. Our framework naturally supports both offline profiling and online dynamic selection, making speculative actions essentially a new application of this model-selection paradigm.
>
> In addition, we also find that small models are great candidates for Speculators. In fact, our E-commerce and HotpotQA environment both uses smaller counterparts of their big Actor models  as Speculators (GPT-5-nano for GPT-5, Gemini-2.5-Flash for Gemini-2.5-Pro, etc.) and obtained roughly 30% speculation accuracy.
>
> **Reference**
> 1. Chen, Lingjiao, Matei Zaharia, and James Zou. "Frugalgpt: How to use large language models while reducing cost and improving performance." arXiv preprint arXiv:2305.05176 (2023).
> 1. Srivatsa, Kv Aditya, Kaushal Maurya, and Ekaterina Kochmar. "Harnessing the power of multiple minds: Lessons learned from LLM routing." In Proceedings of the Fifth Workshop on Insights from Negative Results in NLP, pp. 124-134. 2024.

---

### Official Review · Reviewer_Ktyn · 2025-11-05

**Soundness:** 3
**Presentation:** 3
**Contribution:** 3
**Rating:** 8
**Confidence:** 2

**Summary:**

The paper proposes a new general framework called "Speculative Actions" in which agents predict future states of the world that will come about as a consequence of e.g. the environment, other actors, computation, API calls, and performs API calls based on that prediction.

**Strengths:**

The idea is solid and the experiments convincingly show a speedup.

**Weaknesses:**

No weaknesses

**Questions:**

No questions

---

> ### Author Response · Authors · 2025-11-21
>
> Thank you for taking the time to review our submission and provide positive feedback!

---

### Author Response · Authors · 2025-11-21
**Rebuttal Summary**

We thank all reviewers for their time and effort in reading our work and providing valuable comments.
Several reviewers highlighted the importance of understanding the cost-latency tradeoff, and we fully agree that this is a central and exciting direction. Motivated by these comments, we have added a new Appendix (Appendix C) containing new theoretical and empirical results. Below we summarize our results:

1. **Static breadth-focused speculation (Algorithm 1) and selection of k.**
   For Algorithm 1, we obtained closed-form expressions for the relative latency improvement ratio and relative cost increase ratio. These expressions allow practitioners to tune the number of speculative branches k *offline* by directly assessing this tradeoff.

2. **Dynamic branch selection substantially reduces cost.**
   Reviewer qQio suggested selecting speculative branches (and effectively selecting k) online using a predictor. We develop this idea formally and prove that, under some modeling assumptions, the *optimal* online strategy is a **threshold rule**: continue only when the predicted accuracy of a speculative branch exceeds a computable threshold. Empirically, this policy performs strongly—on the chess environment, it **achieves the same speedups with 40% fewer tokens**.

3. **No exponential blowup in depth-focused speculation.**
   We analyze a depth-focused speculative policy supporting multi-step speculation. We show that the number of parallel branches remains bounded. We also provide a characterization of the resulting cost–latency tradeoff.

4. **Environments that can reduce both cost and latency.**
   Interestingly, cost and latency do not always trade off against each other. In our OS-tuning environment where losslessness is not required, the Speculator accelerates convergence to the optimum so effectively that the system reaches the solution **using only 8% of the original time and 20% of the original token cost**.

5. **No/low cost for self-serving deployments.**
   The discussions above measure cost in terms of *token count*, but in **self-hosted inference** the relevant cost is GPU-hours or fixed procurement costs—not per-token pricing. Under low-traffic conditions with continuous batching, speculative calls become **effectively free** because they utilize otherwise idle GPU compute bandwidth. Moreover, this opens up new directions in scheduling and workload balancing for optimizing latency and throughput in agentic LLM deployments.

We refer the AE and reviewers to our response to Reviewer hwQA for a concise overview, and to the new Appendix C for full details. Taken together, these results provide a clear and flexible understanding of the cost–latency tradeoffs across both breadth- and depth-focused strategies. We find this direction highly promising, and we will incorporate these results and the accompanying discussions into the main text of our camera-ready submission. We thank the reviewer once again for eliciting this valuable discussion.

---

### Meta-Review · Area_Chair_pr62 · 2026-01-07

**Summary:**

1. Two reviewers identified the cost-vs-latency tradeoff as a critical aspect of the proposed method, emphasizing that the additional costs incurred by the method might offset the latency reduction gains it achieves.
2. One reviewer raised concerns about the lack of a clear description of how the Speculator model is selected.
3. Another reviewer primarily criticized the accuracy of formulas and formatting throughout the paper.

**Reviewer Concerns:**

1. Addressed Concerns
  - Reviewer qQio’s concerns regarding the accuracy of the paper’s formatting, figures, and formulas were fully addressed.
  - Reviewer SJo2 highlighted two key gaps: the absence of cost analysis and the lack of clarity on Speculator model selection. The rebuttal resolved these issues by adding a comprehensive cost analysis in the appendix. It also clarified that Speculator model selection is the focus of a separate, independent line of important work, while presenting empirical findings that smaller models are more suitable for use as Speculator models in the current study.

2. Outstanding Concerns

  - Reviewer hwQA’s core concern: Does speculative execution introduce hidden costs that offset latency gains?The rebuttal proposed two solutions: (1) Under scenarios where accuracy loss is acceptable, both cost reduction and latency optimization can be achieved simultaneously. However, this approach violates the core feature of the proposed method—losslessness. (2) Assuming self-deployed agents with idle servers, the additional cost of speculative execution can be avoided by utilizing idle resources. Nevertheless, server idle time is extremely rare in practice, and these idle servers could alternatively be directly allocated to accelerate the actor model itself.Thus, the authors failed to adequately address the critical issue of cost-latency offset, which significantly limits the practical applicability of the proposed method.

**Reviewer Scores:**

- Reviewers SJo2 and hwQA both emphasized the cost of the proposed method as a critical concern. The concern is that  although the authors supplemented cost analysis in the appendix, the results clearly show that costs increase multi-fold with the proposed method. Nevertheless, both reviewers could increase their scores.
- Reviewer qQio focused exclusively on formatting issues, which were fully addressed. Their score could increase.

---

### Decision · Program_Chairs · 2026-01-26

Accept (Oral)